History of Geo- and Space Sciences Discussions Open Access

# History of Kakioka Magnetic Observatory

Ikuko Fujii[1], Shingo Nagamachi[2]

[1]Meteorological College, Kashiwa, 270-, JAPAN
[2]Kakioka Magnetic Observatory, Japan Meteorological Agency, Ishioka, 315-0116, JAPAN

*Correspondence to*: Ikuko Fujii (ifujii@mc-jma.go.jp)

**Abstract.** Kakioka Magnetic Observatory (KMO) was founded in 1913 by Central Meteorological Observatory (CMO, later Japan Meteorological Agency) as a successor of Tokyo Magnetic Observatory. Kakioka was a village 70km north of Tokyo and was selected to escape from tram noise at Tokyo. At first, it was an unmanned observatory only for the geomagnetic field observation. Then, the Great Kanto Earthquake in 1923 changed the fate of KMO because the earthquake severely

damaged CMO at Tokyo and recording papers of KMO were lost. KMO was manned in 1924 and was redesigned as an institute for geophysics rather than geomagnetism. KMO operated a variety of observations such as the atmospheric electric field, the geoelectric field, the seismicity, the air temperature, the wind velocity, the sunspot and solar prominence as well as the geomagnetic field by 1940's. In addition, research activity flourished with the leadership of the first director Shuichi Imamichi. After the World War II was over in 1945, KMO formed a network of observatories in Japan by founding several

branch observatories originally for the geoelectric field observation. Two branch observatories at Memambetsu and Kanoya survived with the geomagnetic field observation added in the International Geophysical Year project (1957-1958). Efforts on development of instruments for the geomagnetic absolute measurement and systems of high sampling recordings in 1950' to 1970's resulted in the development of Kakioka Automatic Standard Magnetometer (KASMMER) system in 1972. KASMMER measured the geomagnetic field every three seconds at the highest standard in the world in a digital form giving

1 minute digital values of the geomagnetic field available. Those system has been updated and the high sampling technology was applied to the geoelectric field observation and the atmospheric electric field observation. Later adding the geomagnetic field observation at Chichijima in 1971, KMO established the unique electric and magnetic observation network at Kakioka, Memambetsu, Kanoya and Chichijima and provided precise and high-speed sampling data (1min, 1sec, and 0.1sec values) by 2001. On the other hand, KMO gradually terminated or automated their observations and reduced their staff for last

several decades following the government's reform policy. The two branch observatories at Memambetsu and Kanoy were unmanned in 2011 and the atmospheric electric field at Memambetsu was terminated at that time. The geoelectric field observations at Kakioka, Memambetsu and Kanoya were terminated in 2021 as well as the atmospheric electric field at Kakioka. KMO focus the geomagnetic observation for now and put efforts to the total force observation at volcanos and the digitization of historic analogue data.

**1 Introduction**

Kakioka Magnetic Observatory (KMO) is one of the longest-serving geomagnetic observatories in the world and is the longest in the East Asia. They have established a unique reputation through their scientific and technological achievements.

It was founded by Central Meteorological Observatory in 1913 at Kakioka where is about 70km north of Tokyo as a successor of Tokyo Magnetic Observatory (Fig.1), because Tokyo Magnetic Observatory was influenced by tram noises.

KMO regularly has observed not only the geomagnetic field but also the atmospheric electric field, geoelectric field, earthquakes and weather elements for more than a century at Kakioka. In addition, KMO has operated Memambetsu, Kanoya and Chichijima observatories for more than 60 years (Fig.1). Most of their digital data are freely downloadable from the data site of KMO's HP (http://www.kakioka-jma.go.jp/obsdata/metadata/en).



**Figure 1: Map of present (red circles) and past (yellow circles) observation sites of Kakioka Magnetic Observatory. The head quarter at Kakioka is marked by a red star. Map was drawn by using Generic Mapping Tools (Wessel and Smith, 1998).**

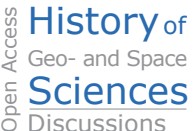

This article describes the history of KMO. Some article already documented the history of KMO (Imamichi, 1938; Japan Meteorological Agency, 1975; Kakioka Magnetic Observatory, 1983; Imamichi, 1983; Kakioka Magnetic Observatory, 1999; Minamoto 2010; Kawamura, 2013). Among them, Kakioka Magnetic Observatory (1983, hereafter KMO1983) is the most comprehensive work and an early part of KMO's history in this article mostly refers to KMO1983. Kakioka Magnetic

Observatory (1999) summarizes the history of KMO's branch observatories (Memambetsu and Kanoya) during 20th century.

**2 Early History**

Early records of the geomagnetic field in Japan can be found in historical documents in 17 – 19th centuries such as field notes by Tadataka Ino who made the first realistic map of Japan in 1800-1816, cruise records (Shinozaki 1938), and notes by local scientists (Imamichi, 1956). Kato (1983) pointed out that Masamune Date, a famous war load in 17th century,

constructed several buildings facing the geomagnetic north. Those were temporal observations and often only the declinations were observed.

Modern-style continuous observations of the geomagnetic field in Japan were started in 1882-1883 in order to anticipate the observation activity of the First International Polar Year (1882-1883). Antoine Henri Becquerel, who represented Japan at The 1st International Electrical Congress in Paris in 1881 and later famously discovered radioactivity, invited Japan to

participate the Polar Year activity and the Japanese government decided to accept it. Then, with some reason, three national institutions separately started the geomagnetic observations in Tokyo in 1882-1883. One of the three institutions is The Geographical Bureau of the Ministry of Interior in cooperation with The Telegraph Bureau of the Ministry of Industry. The Geographical Bureau of the Ministry of Interior had a division that was in charge of observations of weather and the atmospheric electric field. They made a magnetometer by themselves and started the geomagnetic observation at a property

of The Ministry of Industry at Aszabu-Imai chou in Tokyo from March 15, 1883. The annual report of The Geographical Bureau for 1885 described that the observation was conducted every hour for first 6 months and then 8 times a day (KMO1983). A part of the data in 1883 can be seen in Knott and Tanakadate (1888).

The other two observations were conducted by Hydrographic Bureau in the Navy (Hydrographic Bureau, 1883) and Geological Survey of Japan (Geological Survey of Japan, 1884, 1885, 1886). Those observations were transferred and

suspended in several times after the First International Polar Year, but they are still on-going today as magnetic surveys in the ocean by Japan Coast Guard and magnetic surveys on land and continuous observations of the geomagnetic field at 14 sites by Japan Geospatial Information Authority of Japan, respectively.

The reason why the three national institutions responded is unknown. At that time, only 14 years passed since the feudal society governed by the Tokugawa shogunate (Edo era) came to the end in the Meiji Restoration and a new social system in

a constitutional monarchy style (Meiji era) started in 1868. KMO1983 suspected that the newly-born government administration in 1881-1883 was still changing and Ministries were competing each other resulting in no communication or cooperation.

The geomagnetic observation at Aszabu-Imai chou was continued after the First International Polar Year until March 15, 1886. By that time, the observation faced some difficulty. According to The Geographycal Bureau (1886), the room

temperature largely changed giving poor geomagnetic observations. Therefore, a new observatory hut was prepared in 1886 at Kitanomaru where Central Meteorological Observatory, upgraded to an independent agency from a division of The Geographycal Bureau, was situated. Kitanomaru was used to be a part of Edo castle at the city centre of Tokyo (Fig.2). In addition, new instruments were introduced: a Masqual magnetometer was imported from France for the variation measurement in 1884, a declinometer developed by Prof. Tanakadate of Tokyo Imperial University was adapted and a dip

circle magnetometer was imported. However, it took several years to make those changes work properly (e.g., Okada, 1933). In 1896, Dr. Gijiro Honma (Kato), who majored in geomagnetism in Tokyo College of Science, started to work for the

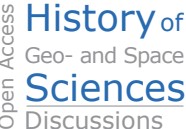

observatory and changed the situation. At last, the variation and absolute measurements of the geomagnetic field were commenced at Kkitanomaru on January 1, 1887. Dr. Honma also prepared the atmospheric electric field measurement at Kitanomaru and its continuous measurement was also conducted there. This is the birth of Tokyo Magnetic Observatory which is the predecessor of Kakioka Magnetic Observatory.

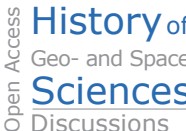

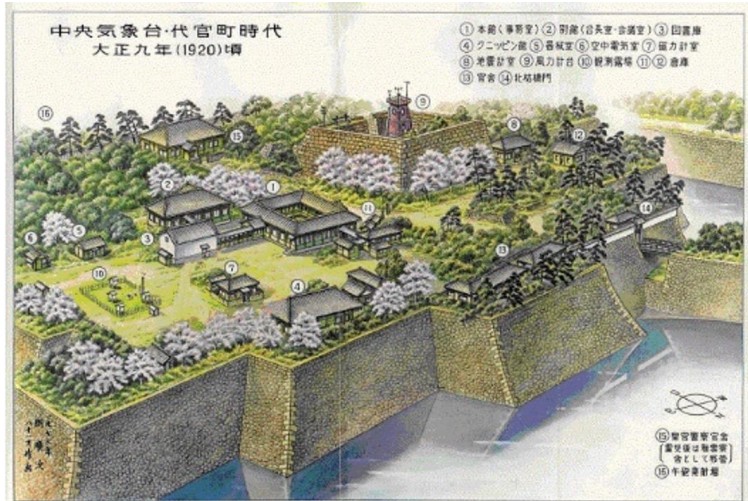

**Figure 2: Central Meteorological Observatory at Kitanomaru of Edo castle. The building numbered as 7 is for the geomagnetic observation and that as 6 is for the atmospheric electric field observation (map supplied by Japan Meteorological Agency).**

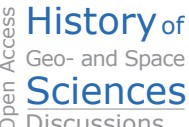

A few officers including Dr.Honma engaged to the geomagnetic observation at Tokyo Magnetic Observatory. Dr. Wasaburo Oishi joined the observatory in 1899 after Dr.Honma had left and installed new instruments such as Eschenhagen variometers, a Wild-Edelman magnetometer, and a Edelman Earth Inductor in 1911. The geomagnetic field data at Tokyo Magnetic Observatory were regularly published on the data books a few years after the observation. Recently, Toya et al.

(2004) corrected obvious mistakes on the data books and made the corrected data available at KMO's homepage (http://www.kakioka-jma.go.jp/en/obsdata/tok_data.html).

### 3 Foundation of Kakioka Magnetic Observatory

As Tokyo grew as a modern city in Meiji era, geomagnetic observations in Tokyo faced an ever-lasting problem; city noise. A city railway network was gradually emerging in 1900's. According to Okada (1937), the observations at Kitanomaru by

Tokyo Magnetic Observatory as well as those at Hongo by Tokyo Imperial University sometimes suffered from noise. At first, Division for city tram of Tokyo city agreed with Central Meteorological Observatory and Tokyo Imperial University to run the railway more than 500m away from their observation sites at Kitanomaru and Hongo. However, a plan of a new tram near Hongo was proposed in 1912 and Central Meteorological Observatory and Tokyo Imperial University decided to move their observation sites by using compensation fee from the city rail.

The two institutions cooperated for the selection of the location for new observations (KMO1983; Terada, 1951; Okada, 1933). They searched places in a vicinity of Tokyo, where was likely to be off future railways. Then, Shigehara (Chiba prefecture) and Kakioka (Ibaraki prefecture) were selected as two candidates. They favored these geological settings of Kakioka. Kakioka is about 13km away from Ishioka Railway Station of Joban Line, situating in a small basin surrounded by Tsukuba mountains at the margin of Kanto plain. The Kakioka basin consists of marine terrace and alluvium along Koise

river. A surface sediment layer over a basalt base rock is thinner than several tens of meters in the basin and the basalt is exposed at Tsukuba mountains (Miyazaki et al., 1996).

Kakioka was a village of about 3000 people in the end of Meiji era (Committee for Compilation of Yasato-cho History, 2005). The downtown was developed along a highway on a hill. Rice fields, forests and small hills surround the downtown hill. Vestiges of human livings in 4th century and a medieval castle are seen today in the center of the downtown.

Central Meteorological Observatory bought an area of 14,691 $m^2$ on the foot of a small hill (Fuji-yama) off the Kakioka downtown with about 1628 yen in 1912. Shortly after, Tokyo Imperial University bought a much larger area including Fuji-yama and it surrounded the property of Central Meteorological Observatory (Fig.3). As KMO expanded their functions later, a part of the property of Tokyo Imperial University was given to or rented by KMO. KMO uses 72,691 $m^2$ today and the two properties are still next to each other. This has been helping KMO because it shields the KMO's observation from artificial

noises.

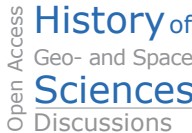

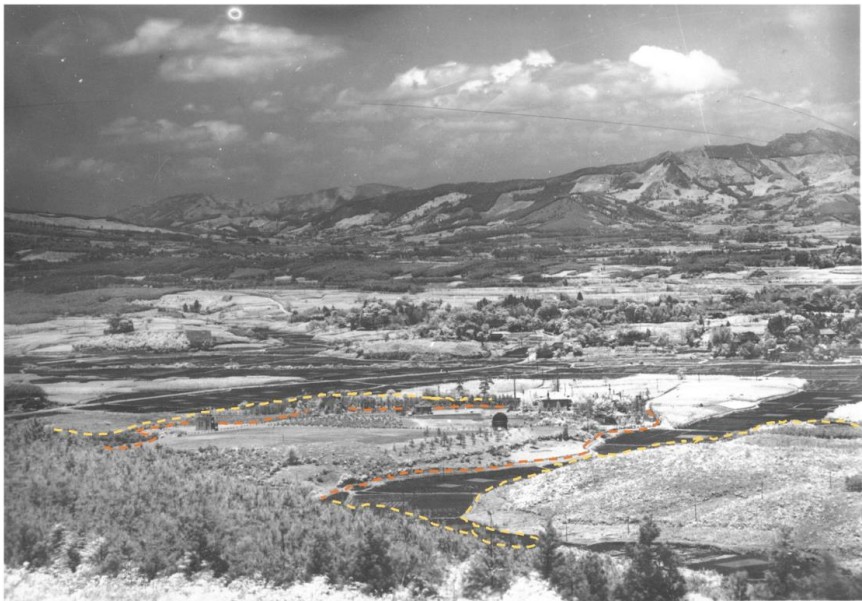

**Figure 3: A view of Kakioka Magnetic Observatory from Fuji-yama in 1933. An orange dotted line indicates the property of the Observatory and a yellow dotted line indicates the property of Tokyo Imperial University (after a photo supplied by Kakioka Magnetic Observatory).**



Central Meteorological Observatory built a dart-covered stone building for the continuous geomagnetic observation (known as 'Ishimuro') and two huts for the geomagnetic absolute observation and the dormitory in 1912 (Fig.4). The Eschenhagen variometers were installed in Ishimuro, and the Wild-Edelman magnetometer was fixed in the absolute hut (Fig.5). Then, the geomagnetic observation at Kakioka was started on January 1, 1913. Weather observations were added in the next year.

5    Thus, KMO was born here.

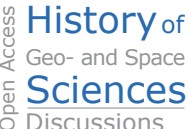

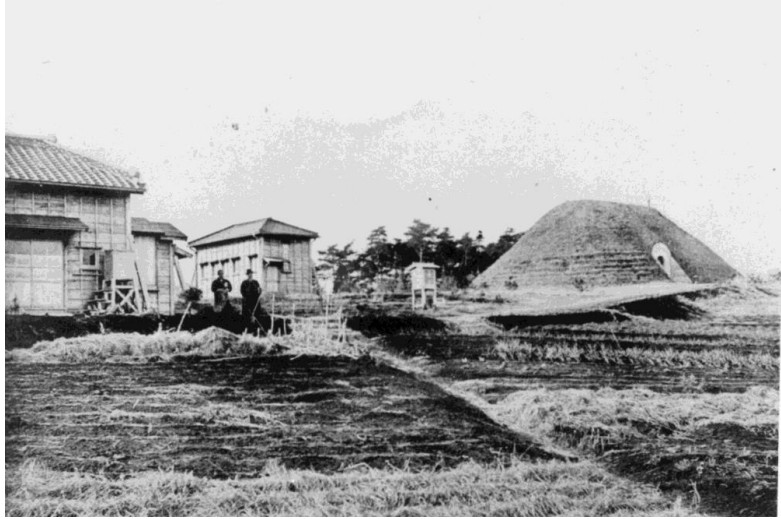

**Figure 4: The 'Ishimuro' variation house (right) and the absolute hut (left). Ishimuro was used till 1980's (photo supplied by Kakioka Magnetic Observatory).**

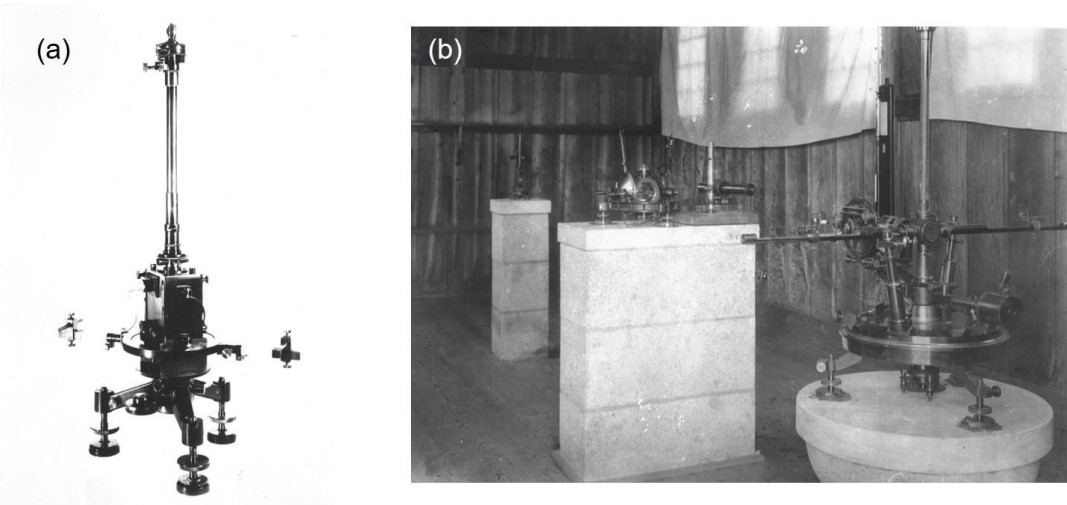

**Figure 5: Instruments for the geomagnetic field measurement. (a) The Eschenhagen variometer for the horizontal component and (b) Wild-Edelmann theodolite and Wild-Edelmann earth inductor (photo supplied by Kakioka Magnetic Observatory).**

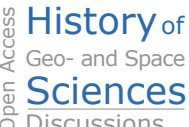

## 4 Reorganization of the observatory

KMO was unmanned at first. Dr.Oishi and other officers of Central Meteorological Observatory visited once a month to do the absolute measurement and a local employee occasionally replaced recording papers. All recording papers and field notes were transferred to the main office of Central Meteorological Observatory in Tokyo where the data were processed and were

published.

When the Great Kanto Earthquake hit Tokyo on September 1, 1923, it couldn't help losing all data observed at Kakioka in 1911 – 1917 by a fire that burned the main office of Central Meteorological Observatory down. The data before 1911 were already distributed as publications. The recording papers and instruments of Tokyo Magnetic Observatory were also burned out. The buildings of KMO were seriously damaged by the earthquake, too.

Dr. Takematsu Okada, the head of Central Meteorological Observatory at that time, decided to make KMO manned so that the data processing, publication and storage could be done there as well as the observation (Fig.6). He also planned to reinforce the observation and research at KMO. The weekly absolute observation was planned and observations of the atmospheric electric field and seismicity were added to the observatory tasks. Later, observations of the geoelectric field and solar surface were started and Dr. Okada described KMO as 'practically, a comprehensive geophysical observatory' (Okada,

1933).

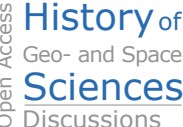

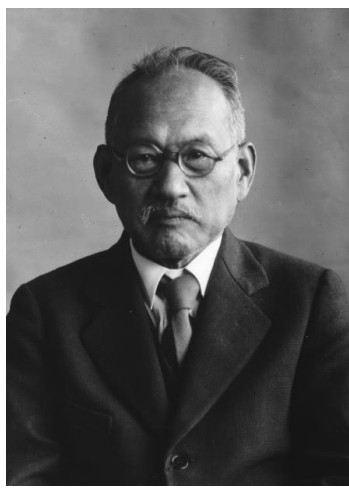

**Figure 6: Dr. Takematsu Okada, the Head of Central Meteorological Observatory from 1923 to 1941 (photo supplied by Japan Meteorological Agency).**

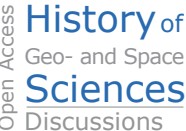

Dr. Syuichi Imamichi was appointed as the first director of a new KMO in December, 1923 (Fig.7). Dr. Imamichi majored in physics at Tokyo College of Science and was in charge of the geomagnetic observation in Central Meteorological Observatory for a while. He offered himself to kick off the newly born observatory and then devoted himself to establish the observatory for about 28 years till he retired from KMO and took a professorship of Tokyo College of Science in 1952.

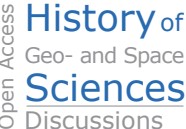

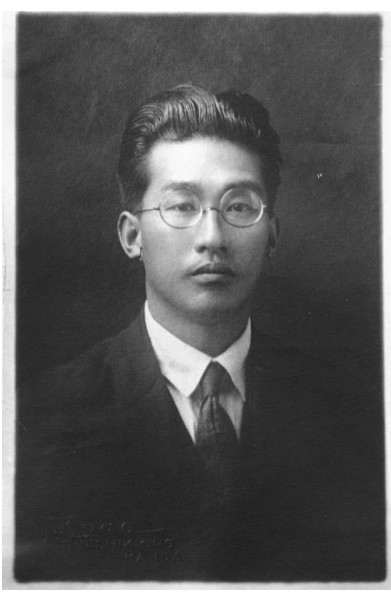

**Figure 7: Dr. Syuichi Imamichi, the first director of Kakioka Magnetic Observatory. He led the Observatory for about 28 years from 1923 to 1952 (photo supplied by Kakioka Magnetic Observatory).**

New buildings at Kakioka were completed in 1924-1925 such as the main office, the absolute measurement hut, the geomagnetic experiment hut, the atmospheric electricity hut, and the variometer measurement hut (Fig.8a-e) and most of them are still used today. Their designs have Spanish or German flavours and were stylish at that time. The main gate of the Observatory whose spherical stones represent the Earth and houses for the staff were also constructed (Fig.9a,b). Director

5 Imamichi and two other officers moved to Kakioka and the geomagnetic observation was resumed in 1924 by using the same instruments as before (Fig.9c). Later in the same year, Prof. Tanakadate reported to a business meeting in the 2nd General Assembly of International Union of Geophysics and Geodesy (IUGG) at Madrid that KMO would serve as the standard observatory in Japan for now.



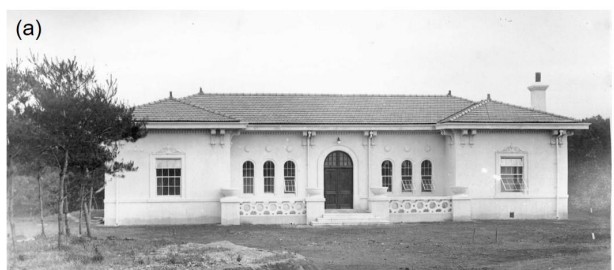
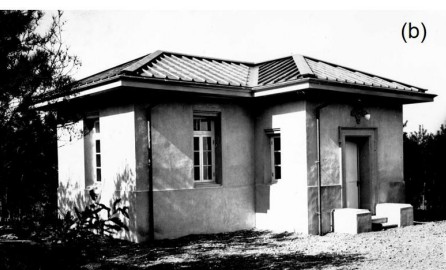

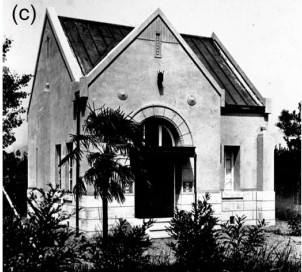
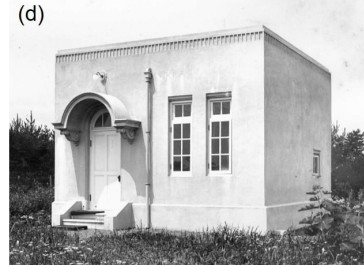
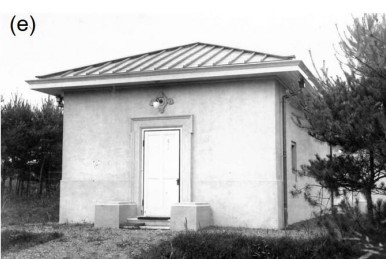

**Figure 8: Buildings constructed in 1924-1925. (a) the main office, (b) the absolute hut, (c)the experiment hut, (d) the atmospheric electricity hut, and (e) the variometer hut are still used today (photo supplied by Kakioka Magnetic Observatory).**

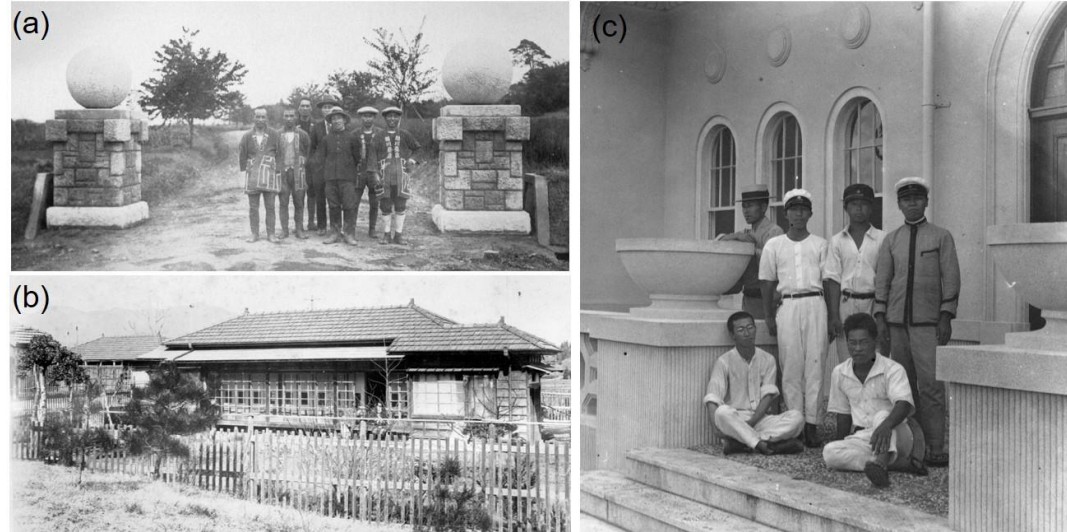

Figure 9: (a) The main gate of the observatory at its completion with craftsmen. Two spherical boulders on the top represent the Earth. (b) Houses for the staff of Kakioka Magnetic Observatory. (c) The observatory staff in August, 1925. The newly manned observatory started with three staff members in 1924 and the staff number gradually increased (photo supplied by Kakioka Magnetic Observatory).

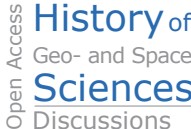

The next task for the Observatory was improvement of the precision in the geomagnetic field observation. The Observatory bought Eshenhagen-Schmitt variometers (Fig.10a), an Askania-Schmitt earth inductor and a Lifler standard clock in 1924 and installed them in new buildings. Then, the old and new instruments ran simultaneously for several years. In the next year, an Indian survey pattern magnetometer (Fig.10b) and a mobile magnetometer by Carl-Vanberg Co. were

5    brought in. Then, a set of Ad Schmitt theodolite (Fig.10c) was bought for the absolute magnetometers. Director Imamichi went to visit Prof. Schmitt in Potzdam and Verlin, Germany in 1927 to learn the operation of the magnetometer and measure its calibration constants.  After he came back to KMO in 1929, the Schmitt theodolite was used in the regular absolute measurements.

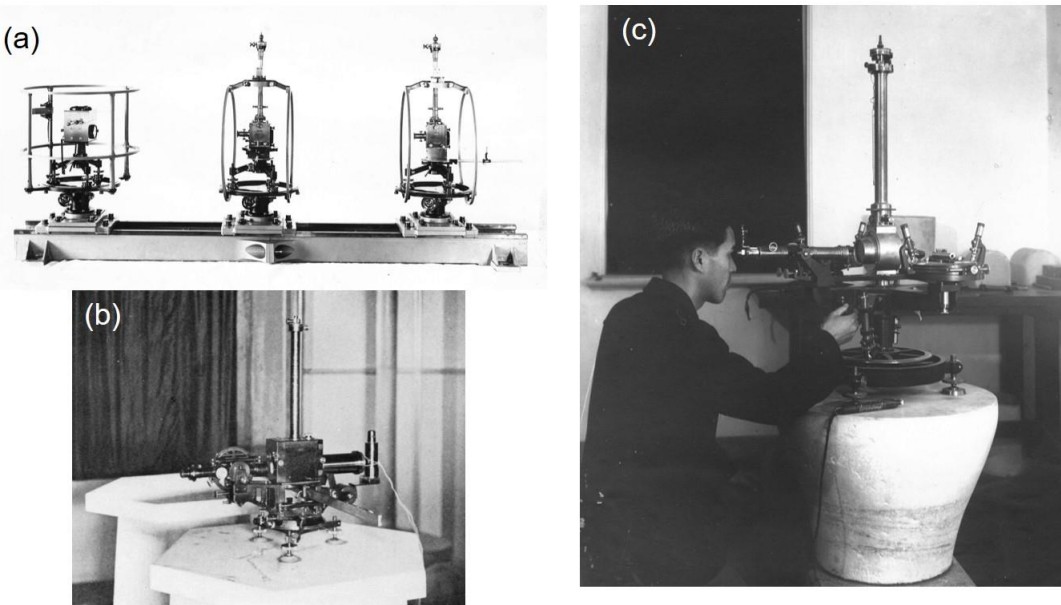

**Figure 10: (a) Eshenhagen-Schmitt variometers, (b) an Indian survey pattern magnetometer, and (c) a Schmitt theodolite for the declination during an observation in 1952 (photo supplied by Kakioka Magnetic Observatory).**

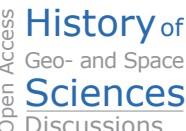

However, the observed geomagnetic field still showed instabilities. Toya et al. (2012) rechecked field notes and recording papers from 1924 to 1946 and found some mistakes in data processing. Corrected data have not been published yet.

In 1927, Mr. Muramoto, an engineer of Hydrographic Bureau, developed a theodolite to observe the declination, inclination and horizontal force and named it the Hydrographic Bureau theodolite (Asao, 2011). It was used for the geomagnetic survey

5    at Hydrographic Bureau, the army, universities and KMO (Fig.11).

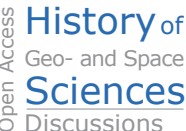

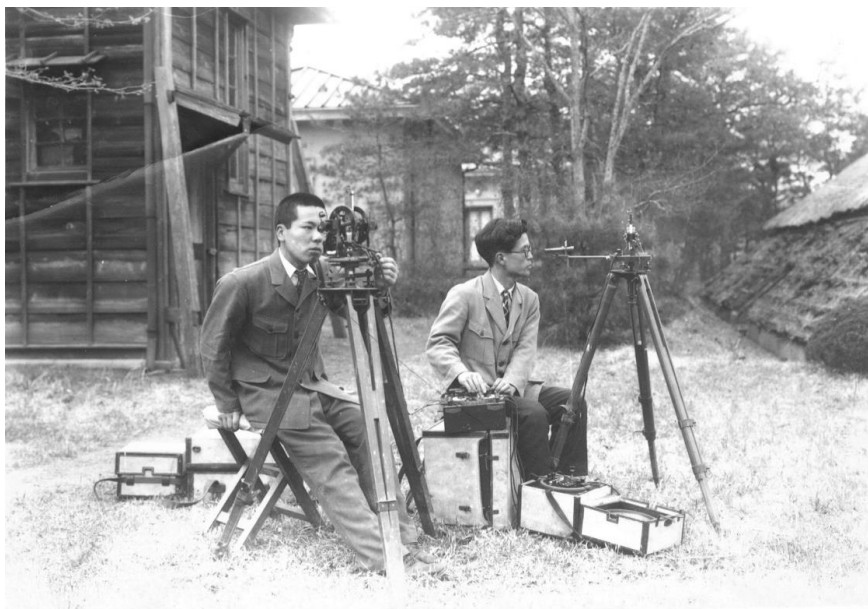

**Figure 11: Hydrographic Bureau theodolite for the horizontal force during an observation in 1947 at Kakioka (photo supplied by Kakioka Magnetic Observatory).**



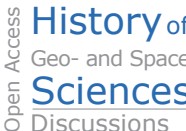

KMO had been expanding in this time period. Director Imamichi made efforts to hire young researchers as the staff following the way the Potzdam Observatory was operated and the number of the staff reached 9 in 1930 from 3 in 1924. New observations were started, too. A seismograph was installed by Dr. Wadachi of Central Meteorological Observatory in 1926 which is a year before he discovered the Wadati-Benioff zone (1927). The observation of the atmospheric electric field

5 was commenced in 1929 with a Kelvin water dropper and a Bendorf's electrometer (Fig.12). The observation of the sunspot and solar prominence was started in 1930 with a 20cm refractor.

Since the First Sino-Japanese War in 1894-1895, Japan had been on the course of seeking the regional dominance of the East Asia. Because of this circumstance, Qingdao Magnetic Observatory in China was consigned to Central Meteorological Observatory in 1922. KMO might help its operation, however details are unknown.

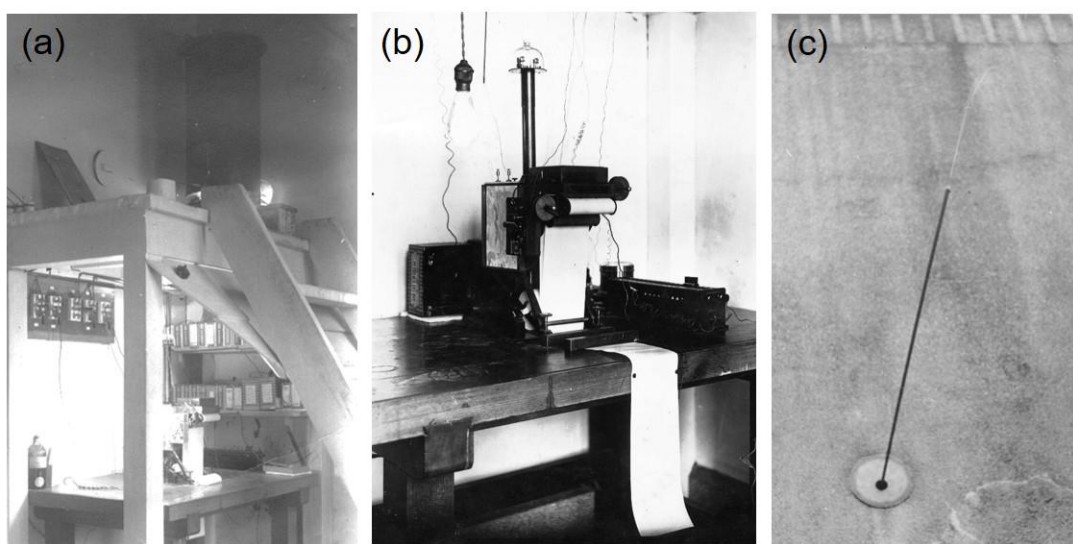

**Figure 12: (a) the observation system of the atmospheric electric field observation in 1937. The Bendorf's electrometer (b) is on a desk in a lower floor and a black water tank of the Kelvin water dropper is aside the wall in an upper floor. A water nozzle (c) extends from the tank through a hole of the wall to the outside (photo supplied by Kakioka Magnetic Observatory).**

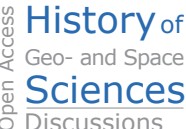

**5 Second polar Year Project and Pacific War**

Central Meteorological Observatory took part in the Second Polar Year project (August, 1932-August, 1933) and added many new observations. That made KMO start the geoelectric field observation in 1932 and a rapid-run observation of the geomagnetic variation. The rain charge, space charge, and atmospheric resistivity were also observed in 1931-1933 (Fig.13).

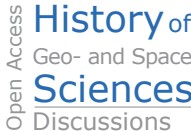

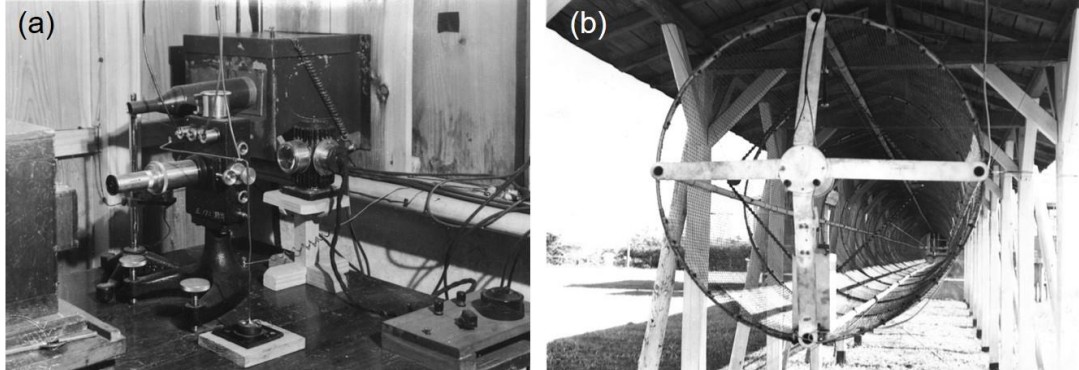

**Figure 13: (a) a space charge meter and (b) a collector part of the air conductivity observation system (photo supplied by Kakioka Magnetic Observatory).**

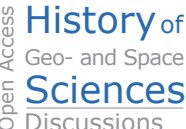

Technical details of the geoelectric field observation at Kakioka are summarised in Yoshimatsu (1957), KMO1983, and Nagamachi et al. (2021). Cupper plates were used as electrodes and they were installed at 3 m deep. The base line length in both the north-south and east-west directions at the property of KMO in 1932 was 100 m (the short base line observation) and then, those outside the KMO property in 1933 were 1120 m in the north-south direction and 1500 m in the east-west

5   direction (the long base line observation). The measurement system consisted of a reflecting galvanometer and a resister (about 10kΩ), and a recording system consisted of a photographic paper which was put on a rotating drum and light sources. The paper feed speed was 15mm/hour for the normal run recording. The geoelectric field observation was run in a similar manner till 1986, however locations of the electrodes were moved several times and electrodes were sometimes changed to carbon rods or pods with lead wires covered with $PbCl_2$. No photographs of the geoelectric field observation instruments in

10   1930's to 1040's, therefore those in 1950's to 1980's are shown in Figure 14.

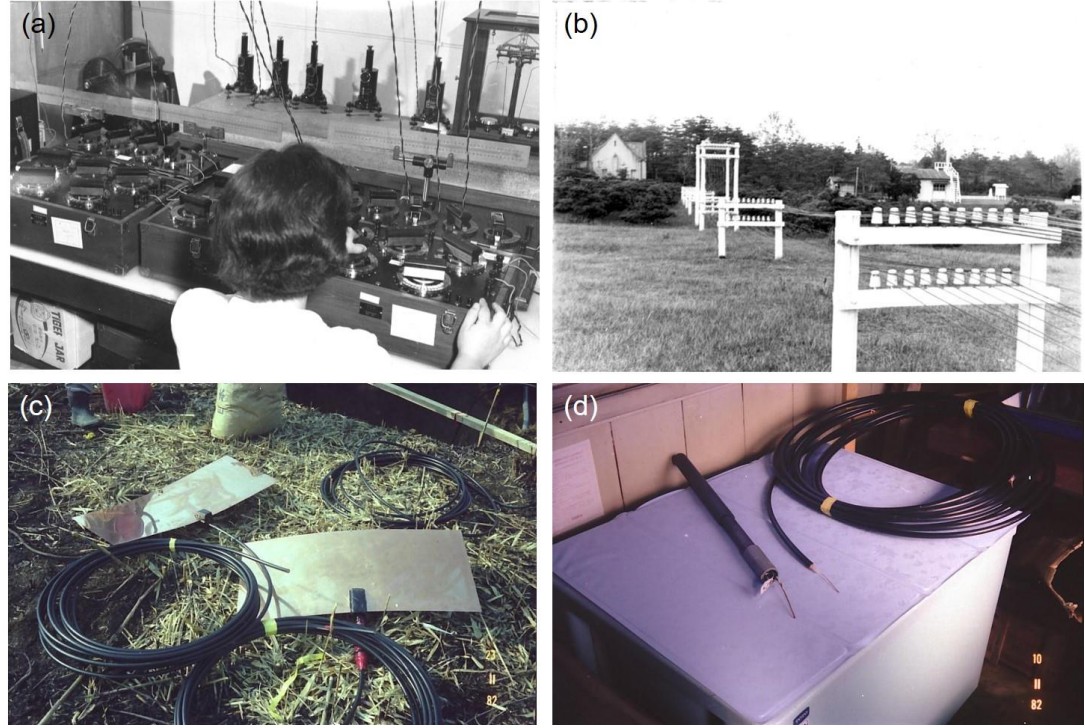

**Figure 14: (a) a reflecting galvanometer and a voltmeter with a 10kΩ resister, (b) cables, (c) a pair of the cupper plate electrodes, and (d) a carbon rod electorode (photo supplied by Kakioka Magnetic Observatory).**

    Then, KMO opened a new magnetic observatory in Yuzhno-Sakhalinsk in Russia in 1932 associated to the Second Polar Year project. This observatory was first named as Toyohara Temporal Magnetic Observatory and is a predecessor of Yuzhno-Sakhalinsk Magnetic Observatory (Fig.15). The geomagnetic and geoelectric fields were continuously observed at Toyohara from 1932 to 1940 and some of those data were published (Central Meteorological Observatory, 1936). The

5    observatory was moved in 1941 and it was renamed as Toyohara Magnetic Observatory. The geomagnetic and geoelectric data at the new place are remained unpublished although copies of some data are stored at KMO.



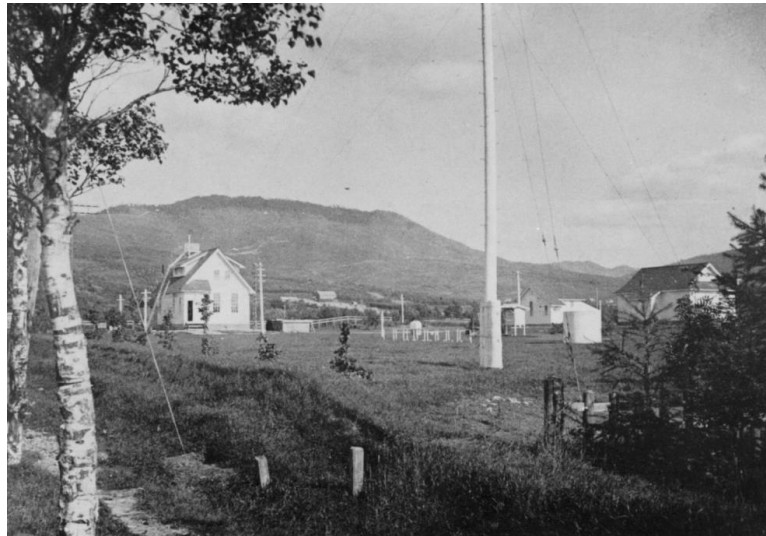

**Figure 15: Toyohara Magnetic Observatory (later known as Yuzhno-Sakhalinsk Magnetic Observatory) (photo supplied by Kakioka Magnetic Observatory).**

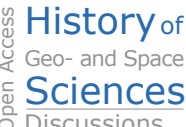

In order to support those activities, the number of the staff was gradually increased from 10 in 1932 to up to 21 in 1941.

The activities for the Second Polar Year project boosted the scientific research at KMO as well as at universities in Japan. For instance, Dr. Misao Hirayama, the second and last director of Toyohara Magnetic Observatory, published one of very first theoretical works on Magnetotellurics, namely the electromagnetic induction inside the Earth (Hirayama, 1934). Dr.

Hisanao Hatakeyama, the first director of Toyohara Temporal Magnetic Observatory and later a head of Central Meteorological Observatory, published his prize-winning paper on the bay and pulsation of the geomagnetic field (Hatakeyama, 1938a,b,c). Director Imamichi revealed that the sudden commencement has a travelling nature by using the geomagnetic fields at Kakioka, Yuzhno-Sakhalinsk, and Qingdao (Imamichi, 1938). Other studies can be seen in many journals, for instance 'Memoirs of the Kakioka Magnetic Observatory' which was established in 1936.

The momentum to promote research activities still continued for a while after the Second Polar Year. Several projects such as the geomagnetic survey in Japan and surrounded areas, the geomagnetic and geoelectric variations under eclipses, and lightning observations started, however some were gradually faded because of a series of wars (the Second Sino-Japanese War in 1937-1945 and the Pacific War in 1941-1945). Then, researches sponsored by military forces were increased. Under that circumstance, Director Imamichi and Tadao Kuboki at KMO kept working on developing new measurement methods to

observe geomagnetic field variations at high frequencies. A Nagaoka-type variometer, XY loop coils (Imamichi, 1954), and a KC-type variometer (Kuboki, 1951) were introduced or developed at Kakioka in 1940's.

Research achievements in Japan of 1940's can be seen in Japan Society for the Promotion of Science (1951) where a variety of studies conducted at KMO such as the magnetic pulsations (KMO1983), Dellinger effect (Imamichi, 1943), magnetic field surveys in Japan (Yumura, 1942), atmospheric electric potential gradients (Hirayama, 1944), electric field

variations due to earthquakes (Yoshimatsu, 1943), light of the night sky (Utsumi, 1943) were reviewed. That suggests that KMO became a leading research Institution in Japan.

In 1940's, essential supplies for living gradually disappeared in shops. Bombing over the Japanese islands often happened, but, fortunately enough, direct attacks to Kakioka were subtle. Many male staff of KMO had to leave the observatory to be conscripted into the army. According to KMO1983, the staff shortage affected even the operation of the regular observations

at KMO although the number of the technical staff was quickly increased up to 57 in 1945. The publication of the data was stopped in 1941. The regular observations were continued mostly by female technical staff members who were mostly hired in 1940's and reached 33% of all staff excluding part timers in 1945 (Fig.16).

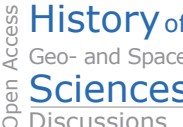

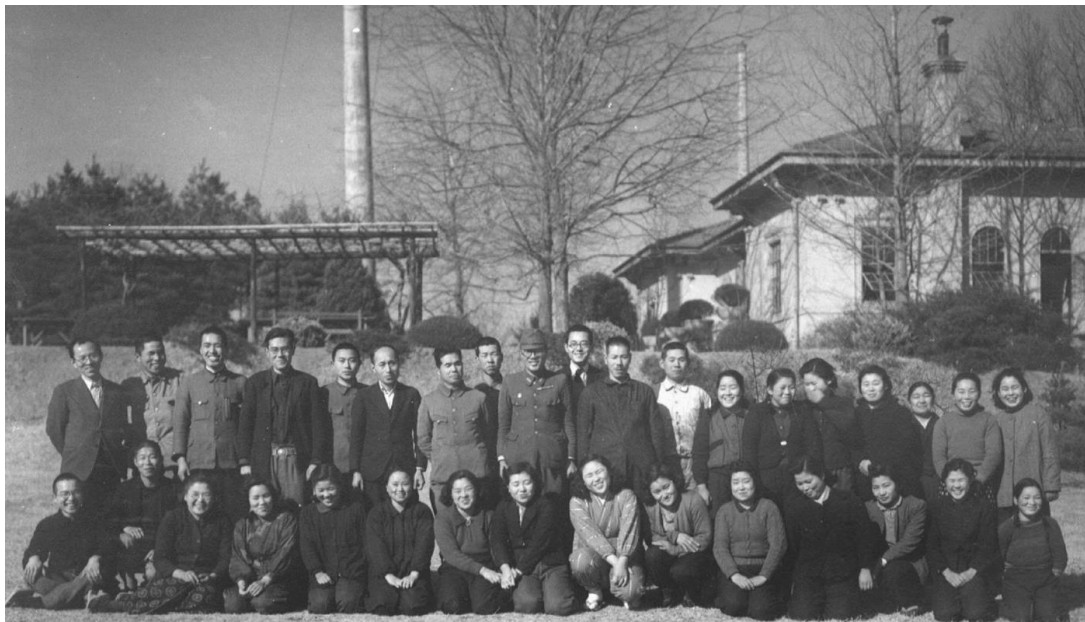

**Figure 16: Staff members at Kakioka in February, 1945 (photo supplied by Kakioka Magnetic Observatory).**

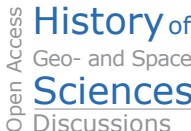

**6 Post War Time**

When the Pacific War was over in1945, the society of Japan faced drastic changes and experienced confusion. New systems of the society were brought in at those days and many of them are still in use at present day. For instance, the Constitution of Japan was established in 1947 being transformed from the Constitution of the Empire of Japan.

Concerning KMO, integration of the three national institutions for the regular geomagnetic field observation in Japan (KMO, Hydrographic Bureau, and Geographical Survey Institute) was discussed. Eventually, it was decided to be unchanged. KMO had to manage to keep the observations going by themselves.

Dr. Imamichi successively led KMO after 1945. The staff number reached 61 in 1947 because some were back from the army or Toyohara. It was reduced to 49 in 1950 following a government policy and the number of the female staff was

gradually decreased.  Dr. Imamichi needed to organize those people and to function KMO as a national standard observatory and research institute.

Two problems that KMO confronted as the standard observatory of the nation were a re-construction of their monitoring network of the regional geomagnetic field and an update of their monitoring instruments.

The observatory at Kakioka was the only working one for KMO in 1945 since Toyohara Magnetic Observatory was

transferred to Soviet Union. On the other hand, the geoelectric field observation was promoted in a new action plan of Central Meteorological Observatory in order to predict earthquakes (Central Meteorological Observatory, 1945). Therefore, KMO opened 5 new observatories in 1946 at Ikutora, Morioka, Haranomachi, Owashi, and Miyakonojyo (Fig.1) and started the geoelectric field observations there. After checking quality of observed data, Ikutora and Miyakonojyo observatories were moved to Memambetsu (MMB) in 1949 and Kanoya (KNY) in 1948, respectively (Fig.1).  The MMB property is used

to be a base for a meteorological troop of the navy, while the KNY one was a private property provided free. Although Morioka and Owashi observatories were closed in 1951, the tentative geoelectric observatories at MMB and KNY were transformed to regular geomagnetic observatories in 1950's (Fig.17). For MMB, the regular observations of the geoelectric field and the atmospheric electric field were started in 1950 with 8 staff members and the geomagnetic field observation was added in 1952.  The regular geoelectric field observation at KNY was started in 1949 with 2 staff members and the

geomagnetic field observation was commenced in 1958. After Haranomachi observatory was closed in 1957, its staff members there were transferred to KNY.

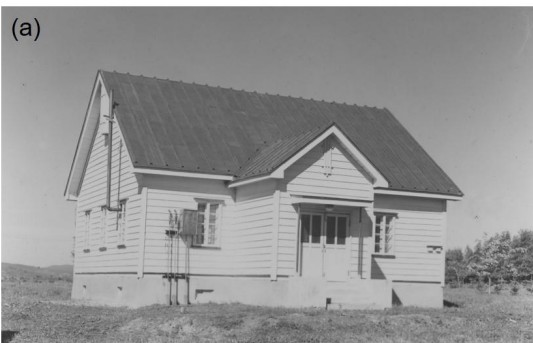
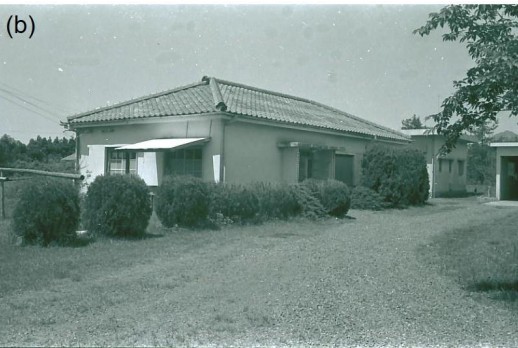

**Figure 17: (a) The absolute measurement house of Memambetsu Magnetic Observatory in 1952. (b) The main office at Kanoya Magnetic Observatory in 1969 (photo supplied by Kakioka Magnetic Observatory).**

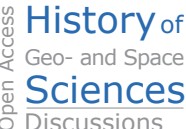

A project to develop new instruments for the geomagnetic field observation was kicked off in 1950's in order to solve a long-suffering problem in the observation accuracy. The fact that KMO was decided to serve as the national institution to authorize magnetometers in 1952 was a booster, too.

Development of a new magnetic theodolite for the absolute measurement at Kakioka was promoted under the cooperation

5   with the science council of Japan and universities from 1952. According to Yoshimatsu (1958), the newly-designed theodolite contained a Helmholtz coil and a search coil in it and resolved 0.5 nT or 3 seconds, which was a high standard at that time. The main theodolite named as A-56 was for the H, Z, D, and I components and another supporting one as H-56 for the H component (Fig.18). Making of A-56 and H-56 required the cooperation of many factories in Japan and it took 4 years to complete. KMO rent a part of the property of The University of Tokyo (former Tokyo Imperial University) and built a

10   new hut for A-56 and H-56 there. Those theodolites started to work in 1958.

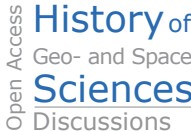
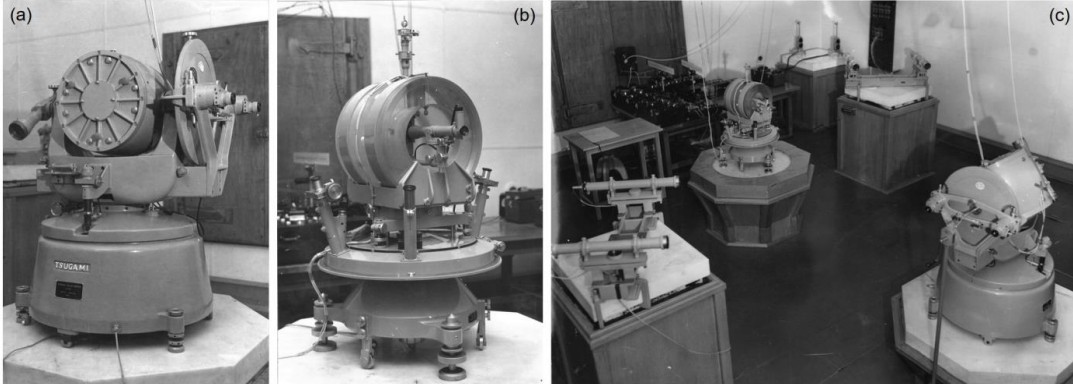

**Figure 18: New theodolites developed by Kakioka Magnetic Observatory. (a) A56 for the horizontal force, the vertical component, the declination and the inclination, (b) H56 for the horizontal force, and (c) the absolute measurement room with A56, H56, and attached equipment such as collimators and a voltmeter (photo supplied by Kakioka Magnetic Observatory).**



At the same time, development of a new variometer was proceeded, too. Based on several experiences of variometers in 1940's, the new variometers were designed and manufactured at KMO (Fig.19). It used a suspended magnet that was fixed with threads of quartz and phosphor bronze. A newly developed magnetic shunt alloy (Kuboki, 1951, 1976) was mounted on the magnet to compensate the temperature effect. First prototypes of the new variometer KM type for the H and D components and KZ type for the Z component was made in 1947 and installation of the final version at KAK, KNY and MMB was completed in 1957 (Fig.20). The KM and KZ type variometers stabilized the geomagnetic variation observation and the data accuracy of the geomagnetic variation met that of the absolute measurement made by A-56 and H-56 (Yanagihara, 1975).



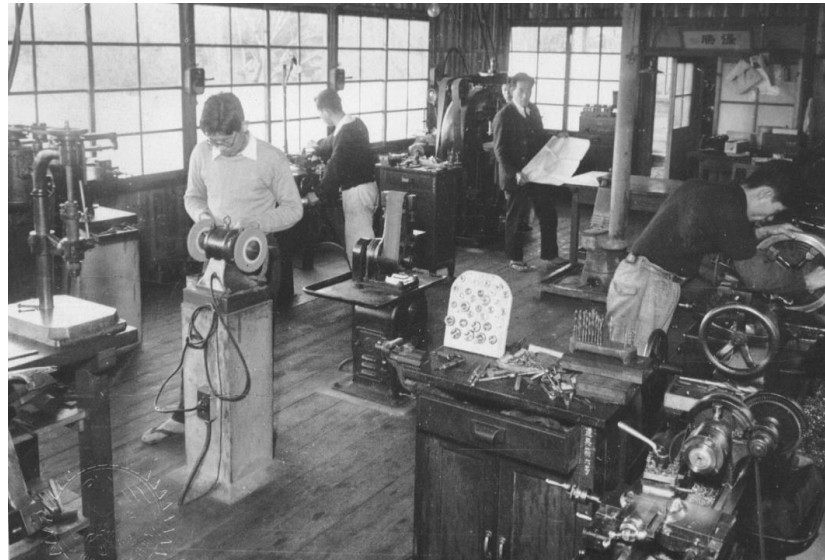

**Figure 19: The craft room at Kakioka Magnetic Observatory (photo supplied by Kakioka Magnetic Observatory).**

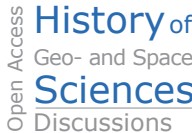

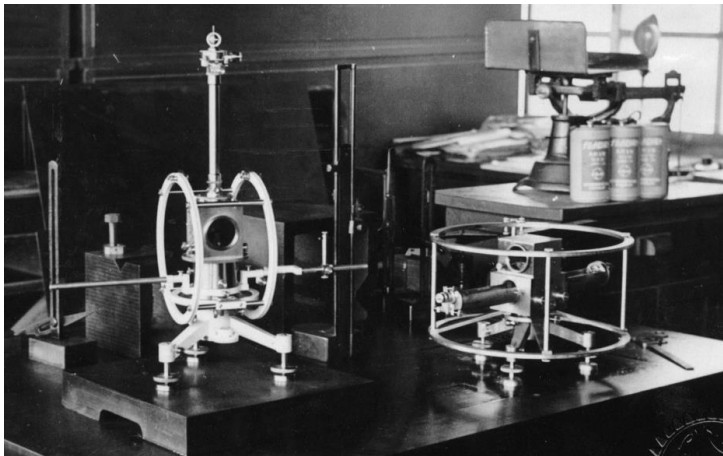

**Figure 20: New variometers designed and manufactured by Kakioka Magnetic Observatory. (left) the KM-type variometer for the horizontal force and the declination, and (right) the KZ-type variometer for the vertical component (photo supplied by Kakioka Magnetic Observatory).**

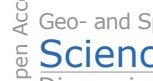

KMO also prompted other observations such as the electric conductivity of the soil by the electric exploration, the ion number in the air in this time period. In addition, K index at KAK was first published in 1951. Publication of the yearbook resumed also in 1951.

Achievements by KMO during this post war time were highly valued among the academics in Japan. KMO was honoured to host the third meeting of the Society of the Geoelectromagnetism of Japan at Kakioka in 1948. The Society was founded in 1947 and the first and second meetings were held at the University of Tokyo (former Tokyo Imperial University) and Kyoto University, respectively. During the meeting at Kakioka, 40 oral presentations were given and Emeritus Prof. Tanakadate's last visit to Kakioka was realised (Fig.21). Ota (1983) recalled that visiting researchers were warmly welcomed by the KMO staff and the villagers and they appreciated that three meals were provided every day because people in city were suffering from food shortage. KMO also hosted the meeting of the Society of the Geoelectromagnetism of Japan in 1952, 1957 and 1962 at Kakioka. Four staff members in total joined the board of councillors for the Society till 1976.

After those efforts to reorganize KMO in a post war confusion, Director Imamichi retired at the age limit in 1952. Takasaburo Yoshimatsu succeeded him. After his retirement from KMO, Dr. Imamichi worked on the geomagnetism at Tokyo Science College and University of Tehran.

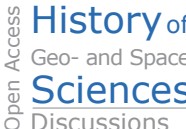

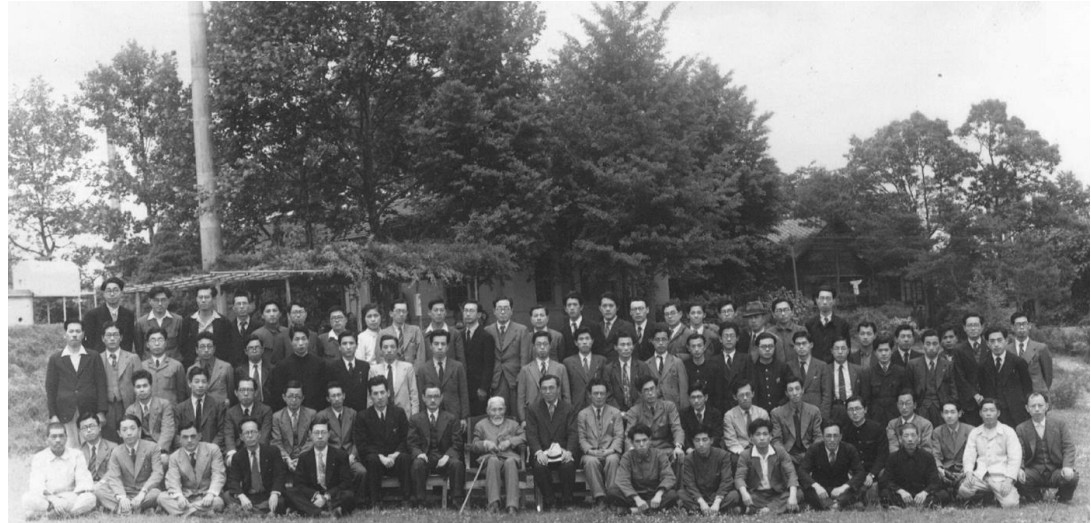

**Figure 21: Attendants of the third meeting of the Society of the Geoelectromagnetism of Japan which was held at Kakioka in 1948. Emeritus Prof. Tanakadate was on a chair with a stick in the front row (photo supplied by Kakioka Magnetic Observatory).**



**7 Train and Law**

Another big topic in 1950's is a train noise at Kakioka. When an Ibaraki section of the Joban train line (a section between Toride and Hirama) was considered to be equipped as a direct-current line to transport large load to Tokyo, a concern arose whether that would cause a noise to the geomagnetic observation at Kakioka. Most of the section locate within 30km from

Kakioka.

Ministry of Transport formed a technical committee to research the noise in 1951 and a test observation of the train noise near a direct-current section was conducted. Then, the ministry organized a meeting to discuss the problem in 1953 among representatives of Ministry of Transport, Japanese National Railways, universities and The Central Meteorological Observatory. The acceptable noise level for KMO was set at 0.3 nT at Kakioka and observational and theoretical studies

indicated that this criterion would be met if the section within 30km from Kakioka was powered by the alternative current. The committee recommended Japanese National Railways to build the Toride-Mito section as an alternative current line (KMO1983).

Thus, the Ibaraki section of the Joban line was powered by the alternative current; the power system was changed between Toride and Fujishiro. The noise at Kakioka was observed and was proved to be less than 0.3nT (Yanagihara, 1977).

A further important step on this issue was taken afterwards. The government and the Ministry of Industry regimented in 1964 and 1965 that commercial electric infrastructure cannot affect geoelectric and geomagnetic field observations by national institutions such as KMO, Geographycal Survey Institute, the Marine Guard, and The National Astronomical Observatory. These unique laws are one of reasons why the geoelectric and geomagnetic observations at Kakioka survive for more than 100 years nearby Tokyo which has been a mega city in the world. However, the protection of KMO keep causing

a strain between KMO and locals, because it is said that it has discouraged more frequent railway services to Tokyo in the alternative current section of Joban line resulting the vicinity of Kakioka remained rural.

**8 Expansion**

The Japanese experienced a quick recovery from the devastation of wars in 1950's and their economy was especially boosted in 1955 – 1973. The national budget for science and technology was quickly increased in late 50's (Kurosawa et al., 1967).

Central Meteorological Observatory was prompted to Japan Meteorological Agency (JMA) in 1956. As for Earth and space sciences, a project on rockets was started at the first time in Japan in Institute of Industrial Science, The University of Tokyo in 1955. The budget on space science and technology increased since. Earthquake Prediction Research Group (1962) published the "blueprint" and it recommended the promotion of KMO's geomagnetic and geoelectric field observations for the prediction study. Then, the First National Earthquake Prediction Programs started in 1965 based on the blueprint. The

program was verified every five years and continued until 1998.

At the same time, a series of international activity on geophysics were conducted, for instance International Geophysical Year (IGY, 1958-1959), International Geophysical Cooperation (IGC, 1959-1963), International Quiet Sun Years (IQSY, 1964-1965), Upper Mantle Project (1964-1966), International Active Sun Years (IASY, 1969-1971), Monitoring of the Sun-Earth Environment (MONSEE, 1972-) and International Magnetosphere Study (1976-1979).

KMO participated those activities and verified their observation to adapt to the activities. Two big changes brought in observations at KMO in 1950's – 1960's would be improvement of the high-speed sampling observation and introduction of new technologies.

Observations of high frequency phenomena were demanded in researches of the solar-terrestrial space physics. The rotation rate of a recording paper was increased and new censors and recording units were developed (e.g., Kuboki, 1951; Kawamura

and Kashiwabara, 1965). Rapid-run observations of the geomagnetic and geoelectric fields were started in 1957 at KAK and MMB and in 1958 at KNY. Ultra rapid-run observations were conducted to observe Pc1 at MMB and KNY in 1964-1966,

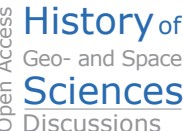

while the ELF observation was introduced at Kakioka in mid-1960's. Then, the ELF and ULF observations were conducted at KAK, KNY and MMB in the IASY period.

New technologies to observe the geomagnetic field were introduced at this time period. A fluxgate magnetometer was used at the first time to observe the geomagnetic field variation aiming at detection of conductivity anomalies in Fukushima in 1964-1965. Then, a long-term observation of the geomagnetic field variation was set at Iwaki (Fig.1) in 1967 as a part of the First National Earthquake Prediction Program. At the same time, KMO developed a new theodolite, MO-P, which consisted of a proton precession magnetometer with two Helmholz coils around it in order to measure the horizontal force and vertical components of the geomagnetic field in the absolute measurement in 1963 (Fig.22). Then, the proton precession magnetometers were added to KAK, KNY and MMB in 1967-1969 to observe secular variations of the geomagnetic total force as a part of the First National Earthquake Prediction Program and then they were used for the absolute measurement, too. The proton precession magnetometer at KAK was transferred to Kitaura in 1982 (Fig.1).

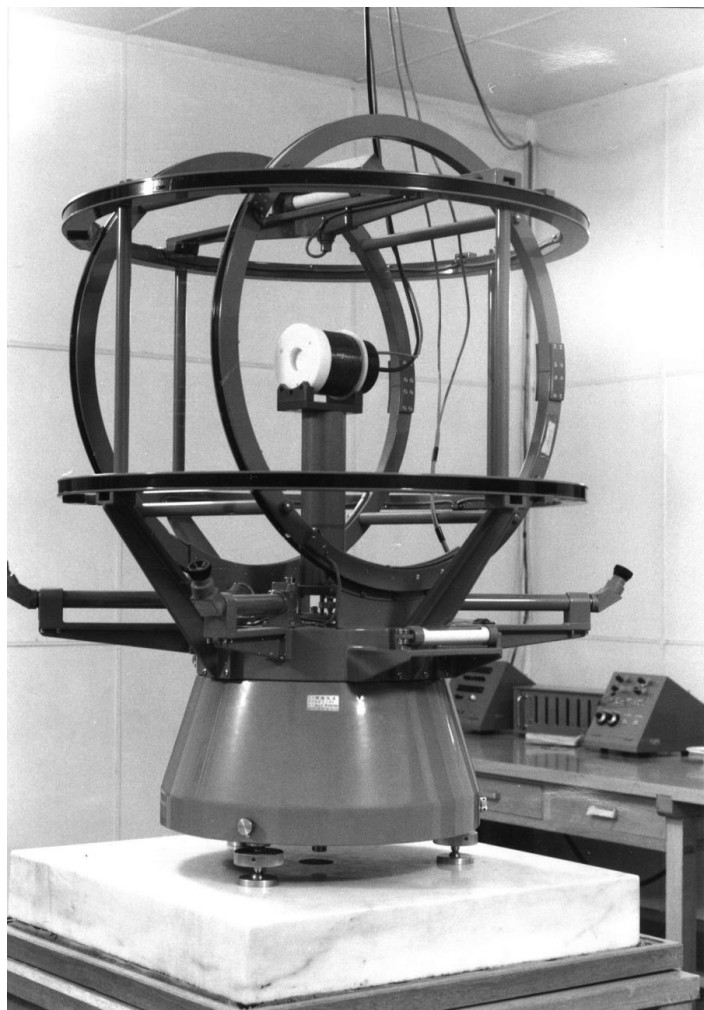

**Figure 22: MO-P theodolite (photo supplied by Kakioka Magnetic Observatory).**



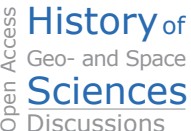

The atmospheric electric field observation was also updated along with those international activities, giving the electrical conductivity observation and the atmospheric electric current observation being added.

Other observations such as the whole sky photography and the geomagnetic vector variation were undertaken in this time period. First photographs of the aurora in Japan were taken on February 11, 1958 at MMB during the aurora observation
5  with a whole sky camera in a cooperative work with Tokyo Astronomical Observatory. These photographs were reanalysed recently to reveal a fan shape of the aurora (Kataoka et al., 2019).

KMO established another geomagnetic observatory at Chichijima (CBI, Fig.1) on an isolated island of the Izu-Bonin arc in 1971 (Fig.23). CBI has been unmanned since the beginning. The geomagnetic field variation has been continuously observed and the absolute measurement has been done by the staff at KAK every a few months.

10  KMO launched a new journal 'Gijyutu Houkoku' which means 'technical report' in English in 1961. Observation and research activity at KMO in that time can be seen in Gijyutu Houkoku as well as Memoir of Kakioka Magnetic Observatory.

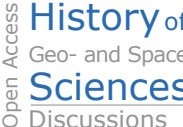

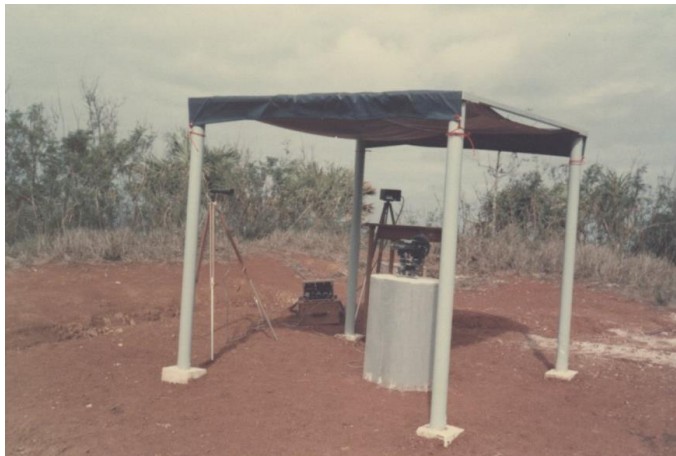

**Figure 23: The pillar for the absolute measurement at Chichijima Magnetic Observatory (photo supplied by Kakioka Magnetic Observatory).**

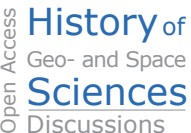

### 8 Kakioka Automatic Standard Magnetometer (KASMMER)

KMO initiated the biggest project in its history in late 1960's. Both of a theodolite for the absolute measurement and a variometer for the continuous measurement had to be updated. A committee was formed in 1968 for a feasibility study to develop a new standard magnetometer system. The team for this project at KMO was led by Dr. Kazuo Yanagihara who had

worked for KMO since 1947 after he had studied physics at the University of Tokyo and served as the 4th director of KMO from 1969 to 1976. Many staff members pointed out that his vision and enthusiasm had driven the project forward (e.g., Kuwashima, 2012; Kawamura, 2013). Dr. Yanagihara was open-minded to use a new technology and was determined to exclude as many noise sources as possible. He recalled his visit to Castlerock Magnetic Observatory in California to see an observation system using brand-new optical pumping magnetometers (Yanagihara, 1983).

The committee finalized an extremely ambitious plan in 1969. In their plan, the geomagnetic field was measured at the highest standard in the world in a digital form by using four optical pumping magnetometers, a newly designed theodolite, a proton precession magnetometer and a computing unit. Not only the accuracy of measured values but also a fine time resolution was ambitious; they planned to measure the geomagnetic field every three seconds and provide 1 minute values averaging 3-second values on the regular basis. Furthermore, the continuous measurement including data processing was

automatically controlled. Buildings were also planned to be newly constructed to reduce observation noises. The whole observation system was named as Kakioka Automatic Standard Magnetometer (KASMMER).

Thus, the 200-million-yen project to construct the observation system of the geomagnetic field in a highest standard was kicked off in 1970. The new theodolite, named as DI72, was consisted of a Helmholtz coil for a rough compensation and a search coil for a fine compensation and measured both the declination and inclination simultaneously by manually rotating

the coils with the resolution of one arc second. A proton precession magnetometer and DI72 formed an apparatus for the absolute measurement (Fig.24).

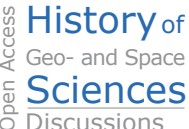

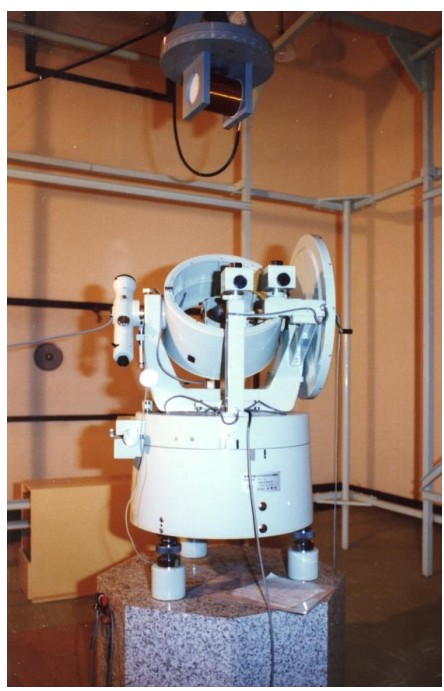

**Figure 24: Newly developed theodolite DI72 beneath a proton processor magnetometer (photo supplied by Kakioka Magnetic Observatory).**

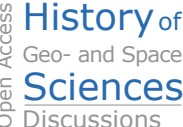

The continuous measurements were done by three optical pumping magnetometers with Helmholtz coils around them for three vector components and one optical pumping magnetometer for the total force (Fig.25). The magnetometers measured the geomagnetic field every 3 seconds with the resolution of 0.1 nT.

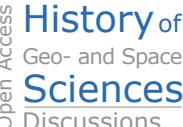
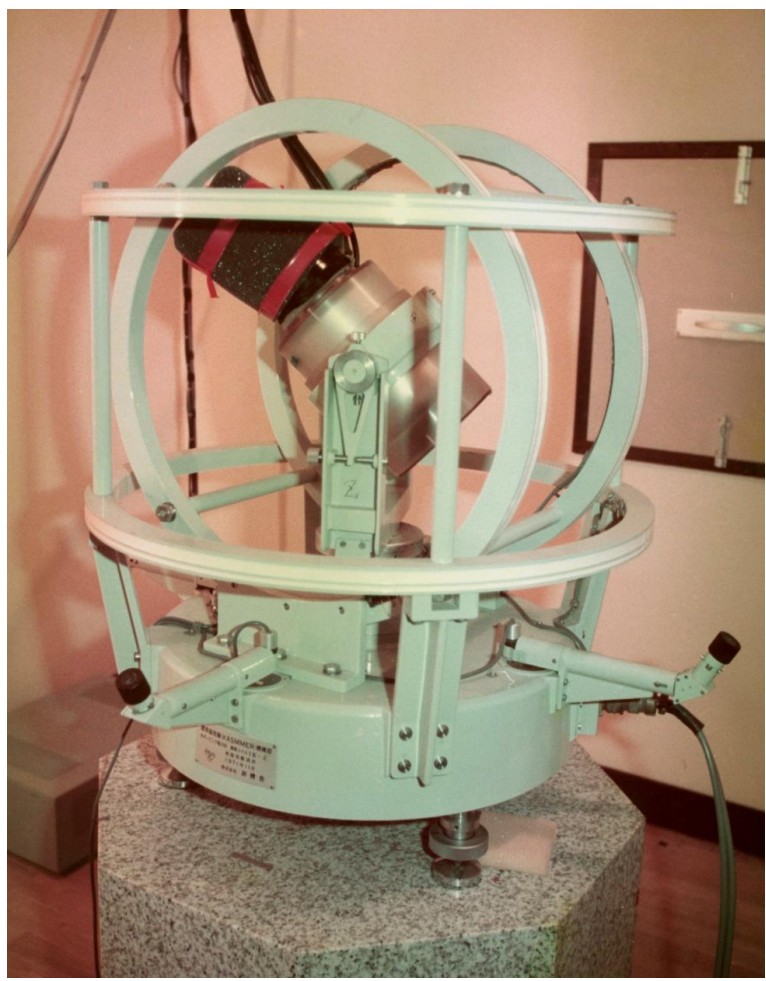

**Figure 25: optical pumping magnetometer with a set of Helmholtz coils surrounded (photo supplied by Kakioka Magnetic Observatory).**

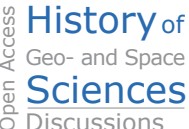

Those magnetometers were connected to the computing unit away from the observation area. The computing unit consisted of two HITAC-6 computers, a storage, a MT tape recorder, typewriters and so on (Fig.26). Observed data by the optical pumping magnetometers were transferred to the computer in real time and the data of the absolute measurement were manually inputted. Those data were automatically and manually processed.

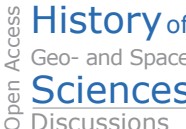

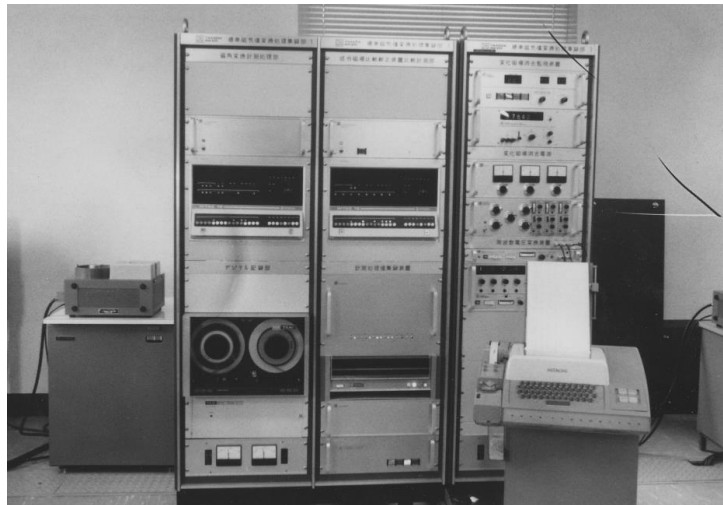

**Figure 26: The computing unit of KASMMER (photo supplied by Kakioka Magnetic Observatory).**



The area of about 13,000 m² at the southern end of the KMO property was levelled to construct several huts for the new magnetometers (Fig.27a). They had to be nonmagnetic and be separated each other so that magnetometers in the huts would not interfere each other. The DI72 and proton precession magnetometer were installed in the biggest hut which was air-conditioned and raised-floor-style in order to reduce the effect of the crustal magnetization. Two observation pillars were 3.6-m tall granite rocks brought from Inada, north of Mt. Tsukuba and were mostly buried into the ground for the stability (Fig.27b). On the other hand, each optical pumping magnetometer occupied a small hut in north of the absolute measurement hut. Those small huts had lower floors compared to the absolute measurement hut and observation pillars were just placed on the base not to disturb the crustal magnetization.

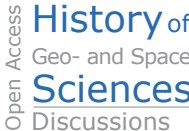

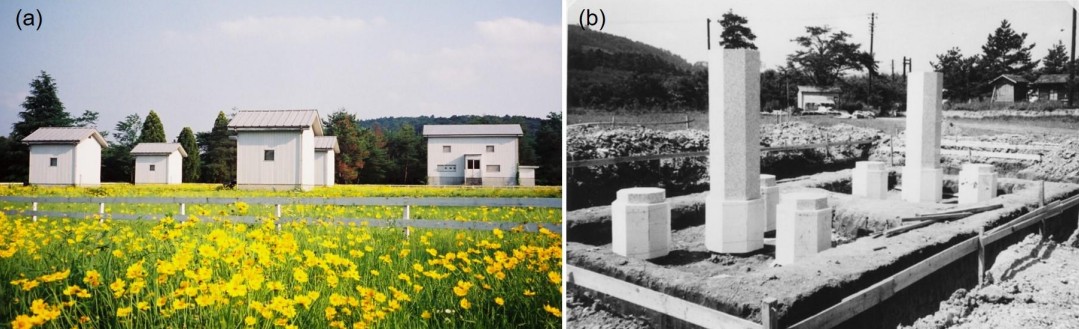

**Figure 27: Huts where KASMMER instruments are installed. (a) The biggest hut is for the DI-72 and four small huts are for optical pumping magnetometers. (b) Two 3.6-m pillars for theodolites were installed in the process of constructing the biggest hut ((a) photo taken by Ikuko Fujii and (b) photo supplied by Kakioka Magnetic Observatory).**



KASMMER was completed in 1972 and succeeded the old system by 1973 making 1 minute values of the geomagnetic field at KAK available. KASMMER's specifications and test results were presented by Yanagihara et al. (1973). They claimed that the continuous observation and the theodolite observation had accuracies of 0.1nT and 1 minute, respectively. Thus, the geomagnetic field observation at KMO reached the highest level.

The absolute observation system at MMB was updated in 1975 following the success of KASMMER (Mizuno et al., 1987). A theodolite DI-75 which is similar to DI-72 and a vector proton precession magnetometer MOP-75 which consisted of a proton precession magnetometer with two Helmholz coils were introduced. MOP-75 was able to measure the H, Z and F components of the geomagnetic field with the accuracy of 0.2 nT. At the same time, the sensor of the atmospheric electric field observation at MMB was changed from a radioactive collector to a field mill.

**9 Post KASMMER**

The Nixon shock in 1971 and oil shock in 1973 caused economic recessions in Japan and then the economy turned booming. KMO suffered from staff cuts and obtained no mega projects, however KMO launched several services and projects in this period by using the data and technologies acquired through the KASMMER project.

The World Data Centre for Geomagnetism, Kyoto started to publish the Dst index in 1973 by using the geomagnetic field
at Kakioka as well as a few other observatories in the world. The geomagnetic field at MMB was included to compute the Kn index in 1968, while that at KNY was used to detect sudden commencements of magnetic storms in 1975 (e.g., Mayaud, 1980).

KMO proceeded for a service of high-speed sampling values in 1980's. The computing unit of KASMMER was updated from HITAC-6 to HAITAC-10 in 1981. That enabled KMO to store 3-second digital values of the geomagnetic field at KAK
in 1980 and then 1-second digital values in 1983 resulting that 1-min values were averaged values not instant values. The 3 second values were published for the first time in the 1983 yearbook and the 1 second value has been published in the yearbooks since 1984. Observation systems of the geomagnetic field were updated in 1979 at MMB, in 1980 at KNY and in 1989 at CBI to make the 1 minute value averaged. Then, updates of recording systems for the geoelectric field were followed at KAK, KNY, and MMB and the 1-min value of the geoelectric field was published at KAK, KNY, and MMB in 1987.
Thus, an unprecedented data set of the geomagnetic and geoelectric fields with the high sampling rate at KAK, KNY, and MMB was made available in 1980's.

Research activity had been promoted in KMO in this period. For instance, the total force observation at volcanos were commenced in 1970's aiming to study geomagnetic variations caused by volcanic activity. Based on the proposal by the Science Council of Japan, the KNY staff started the repeat observation at Sakurajima and Aso volcanos in 1977-78. Then,
the KAK staff started the repeat observation at Kusatsu-Shirane volcano in 1976 and the MMB staff started the repeat observation at Meakandake volcano in 1977. The observations at Aso, Kusatsu-Shirane and Meakandake volcanos have been expanded and are still on-going at present giving unique data sets to indicate long term trends of thermal activity in volcanos (e.g., Takahashi and Fujii, 2014; Takahashi et al, 2018).

KMO conducted many cooperative research projects with universities and research institutions. For instance, the three
vector components and total force of the geomagnetic field and the two horizontal components of the geoelectric field were observed at Omaezaki and Matuzaki in Tokai region aiming at the earthquake prediction in 1979 and geophysical exploration was added in 1982.

Participation to the Japanese Antarctica Research Expedition started in 1968. Dr. Goro Kondo observed the atmospheric electric field in Antarctica as well as the geomagnetic field as a member of the 10th Japanese Antarctica Research
Expedition (Kondo, 1971). Since then, a KMO staff has joined the Expedition every several years for the geomagnetic field observation and conducted researches on such as the atmospheric electric field, aurora, or night sky as well.



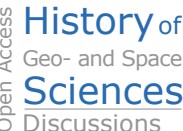

Under the blooming economy, many villages and towns had been urbanized or suburbanized in Japan giving frictions between neighbours and KMO. The noise level at KNY became worse in late 70's and that is an on-going problem at present. KMO built a reference observation point, Haraigawa, 2 km away from KNY in 1988 in order to compare the geomagnetic field variations at the two sites (Sakai et al., 1989). The long baseline observation of the geoelectric field was terminated in

1988 at KAK, because the growing population of Kakioka made KMO difficult to keep a large scale observation system outside the KMO property.

However, the biggest conflict happened in the beginning of 1980's. The observation at KAK faced a local request to move in order to reinforce the railway service to Tokyo. Conferences were held in 1982-83 by the local government of Ibaraki prefecture, the Ministry of Transport, Japanese National Railways, academics and JMA to consider a possibility that the

observation at KAK was moved away from the Joban railway line. At the same time, JMA made a committee to discuss necessary duties of KMO and the Society of the Geoelectromagnetism of Japan examined possible influences of the move. The move was not realized in the end because the estimated cost of the KAK move was too expensive for any organization to afford. However, it was concluded that the KAK move itself was scientifically possible (Ibaraki prefecture, 1983; 1986).

After those high and low times, KMO celebrated the centennial anniversary of the geomagnetic field observation in Japan

and 70th anniversary of the geomagnetic field observation at Kakioka in 1983 (Kawamura, 2013). The book 'Centennial history of geomagnetic observations in Japan' (KMO1983) was published by KMO and the ceremony was held at Kakioka in March, 1983 (Fig.28).

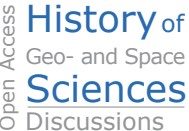

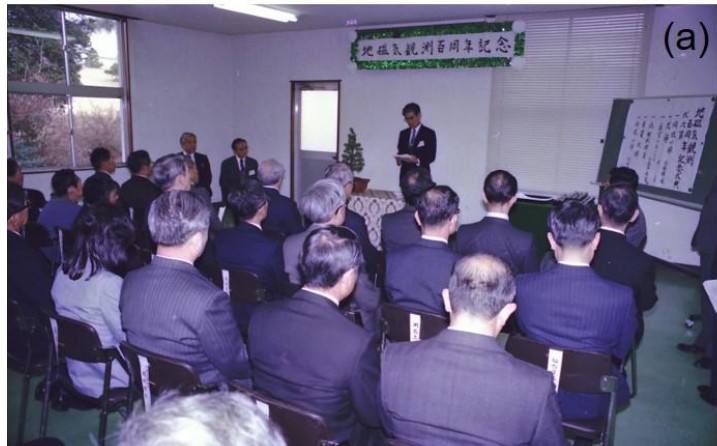

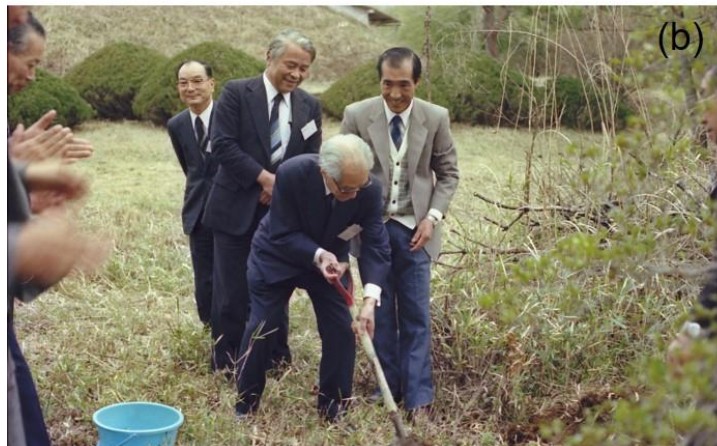

**Figure 28: The ceremony of the centennial anniversary of the geomagnetic field observation in Japan and 70th anniversary of the geomagnetic field observation at Kakioka held in March, 1983. (a) A speech by Dr.Masuzawa, Director General of Japan Meteorological Agency and (b) memorial tree planting by Dr.Imamichi, the first director of Kakioka Magnetic Observatory (photo supplied by Kakioka Magnetic Observatory).**

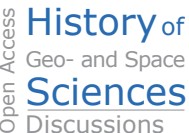

### 10 Japan Meteorological Agency and Kakioka Magnetic Observatory

KMO has been an institute of JMA since its foundation, however JMA (former Central Meteorological Observatory) were transferred from several ministries and JMA's object was changed depending on the ministry to which it belonged.

Central Meteorological Observatory was founded by Ministry of Interior in 1875, then was placed under Ministry of Education from 1895 to 1942, and was transferred to Ministry of Transport in 1943 (Japan Meteorological Agency, 1975). Ministry of Transport was reformed to Ministry of Land, Infrastructure and Transport in 2001. Since Ministry of Education was in charge of administration related to scientific research, Central Meteorological Observatory was active in researches as well as weather forecasting service. Under Ministry of Transport, Central Meteorological Observatory (JMA after 1956) gradually gave priority to service to the public (Wakabayashi, 2019). Based on Meteorological Service Act enacted in 1956, JMA had been establishing its administrative system since 1950's and had been expanding or elaborating their services such that they started numerical weather predictions in 1959, launched a nowcast service in 1988 and made the seismology and volcanology department in 1984 to monitor seismic activity in Japan more intensively. By 1980's JMA grew to a huge organization with more than 6,000 employees and conducted observations, analysis and forecasting/warning services of climate, oceanography, seismology, volcanology etc. Most of JMA's observatories and institutes had no duties related to electromagnetism.

The Japanese government started to reform the administration system at the beginning of 1980's for fiscal reconstruction (e.g., Second Provisional Commission for Administrative Reform, 1983). As part of that action, JMA made several significant changes in its administration in 1980's to 1990's and basically those system has been working since. For instance, JMA defined KMO as an Auxiliary Organ in 1984 together with Meteorological Research Institute, Meteorological Satellite Centre, Aerological Observatory, Matsushiro Seismological Observatory, and Meteorological College. In addition, the human resource planning was centralized so that the director of KMO and new employees are basically appointed by the headquarter in Tokyo. As JMA employees usually move to other positions every a few years, KMO has been involved into this system as one of JMA's institutions.

JMA's transformation influenced KMO in various ways. For instance, KMO had five directors (excluding two substitutes during the absence of Dr.Imamichi, the fist director) for 60 years from 1923 to 1983, while they had 19 directors for 38 years since Dr. Makoto Kawamura, 5th director, left KMO in April, 1983. Many directors after 1983 had never worked for KMO before they became the directors nor experienced works related to geoelectromagnetism. At the same time, majority of division heads of KMO have been new to the KMO works. Before 1983, directors and division heads were often selected among the KMO staff who experienced works of KMO for many years and did researches on geoelectromagnetism. Namely, the way of the leadership changed in 1983.

On the other hand, in 1970's, JMA examined possibilities to centralize the research activity into Meteorological Research Institute so that the other Auxiliary Organs including KMO could focus on observations and public services. No drastic changes happened in the end and those Auxiliary Organs keep their research duties at present. However, several changes have been introduced to restrict research activities there. For instance, only researchers of Meteorological Research Institute qualify to apply Grants-in-Aid for Scientific Research by Japan Society for the Promotion of Science which is one of the biggest research grants in Japan. Researchers of the other Auxiliary Organs could apply some other grants but opportunities were gradually shrunk.

KMO obtained a research grant from Funds for integrated promotion of social-system reform and research and development to run the observations of the geomagnetic and geoelectric fields in Tokai region in 1979 and maintained the observation until 1996. Then, since 1996, KMO ran a 5-year JMA project to observe the geoelectric field with long-baselines in Awaji-shima island in order to investigate seisimo-geoelectric signals originated from aftershocks of the Great Hanshin-Awaji Earthquake in 1995 (Kakioka Magnetic Observatory, 2002). After that, KMO has joined several research projects and played supporting roles. For instance, KMO cooperated with Earthquake Research Institute, the University of Tokyo in



Ocean Hemisphere Network Project from 1996 to help the construction of an absolute measurement point of the geomagnetic field in Christmas island in 1997 and to conduct the absolute measurements of the geomagnetic field at Ponape, Marcus, etc.

Under increasing pressure to enhance public services and narrow research opportunities, research staff gradually left KMO
in 1980's to 1990's, while skilled technical staff tended to stay at KMO because the skills of the geoelectromagntic observations were not used at other JMA institutes. Thus, the personal tradition that employed young academics and grew them to leaders, which Dr. Imamichi, the first director, introduced, finished playing a role by the end of 20th century.

Under such a changing circumstance, KMO kept updating their observations through 1980's to 1990's and technologies introduced in this time still work there. The observation system for the geomagnetic field variation was updated at KAK in
1989-92. The fluxgate magnetometer was introduced in 1989 for higher frequency observations and 4 optical pumping magnetometers were replaced with 4 Overhauser magnetometers in 1990-1992 serving as substitutes (Tsunomura et al., 1994). The fluxgate magnetometer was installed in an air-conditioned room at 5 m deep in order to keep the air temperature as constant as possible (Fig.29).  Then, a level meter was attached in 1997 to monitor ground movements due to earthquakes and so on (Shigeno et al., 1999). The 1-second values of the geomagnetic field variation at KAK observed by the fluxgate
magnetometer were started to be published in 1993, and then the 0.1 second values were started to be published in 1997. Fluxgate magnetometers were installed at KNY in 1995 and at MMB in 1996 (Owada et al., 1998). The absolute measurement at KNY was also updated in 1995 as a new absolute hut was built and a DI-Flux magnetometer was introduced following using a DI-Flux magnetometer for the absolute measurement at MMB in 1992.

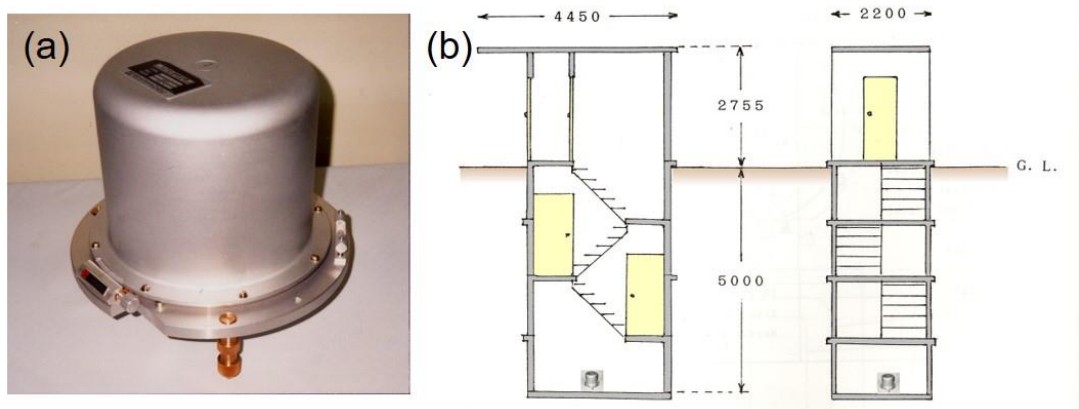

**Figure 29: (a) FM90, the fluxgate magnetometer with level meters. (b) Cross sections of the observation hut which was constructed for the fluxgate magnetometer. The Fluxgate magnetometer is installed at the depth of 5 m (photo and document supplied by Kakioka Magnetic Observatory).**

KMO also updated their geoelectric field observation to make it with high sampling rates. The 0.1 second values of the geoelectric field were made available at 1996-1997 at KAK, MMB, and KNY and the 1 second values were made public in 2000-2001. Mori (1985;1987) developed a new method to measure geoelectric potential differences over long distances with telephone cables and later those works led to a nationwide research project "Network MT" (e.g., Uyeshima et al., 2001).

Internationally, KAK became a member observatory of International Real-time Magnetic Observatory Network (INTERMAGNET) in 1992 and MMB followed next year. Participation of KNY was delayed till 2002.

**11 Kakioka Magnetic Observatory in the 21th century**

The circumstance around KMO has been tough in the 21th century. It has been facing a continuous pressure of small budget and there have been very limited opportunities to hire people who major the geoelectromagnetism. However, KMO
has been searching their way to fit with the time.

KMO reformed a way of public communication and deliverable publication at the beginning of 2000's. It established The Observatory News in 2002. Two journals, Memoirs of the Kakioka Magnetic Observatory and Gijyutsu houkoku, were integrated into Technical Report of the Kakioka Magnetic Observatory in 2003. The CD-ROM yearbook with an original plotting software was published for the first time in 2003.

KMO hosted the 11th IAGA Workshop on Geomagnetic Observatory Instruments, Data Acquisition and Processing on Nov.9-17, 2004 (Okada et al., 2005). The measurement session was held at Kakioka and the scientific session was held at Tsukuba, a nearby city where Meteorological Research Institute of JMA is situated (Fig.30). 147 Participants came from 32 countries.

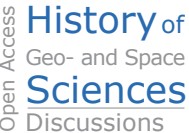

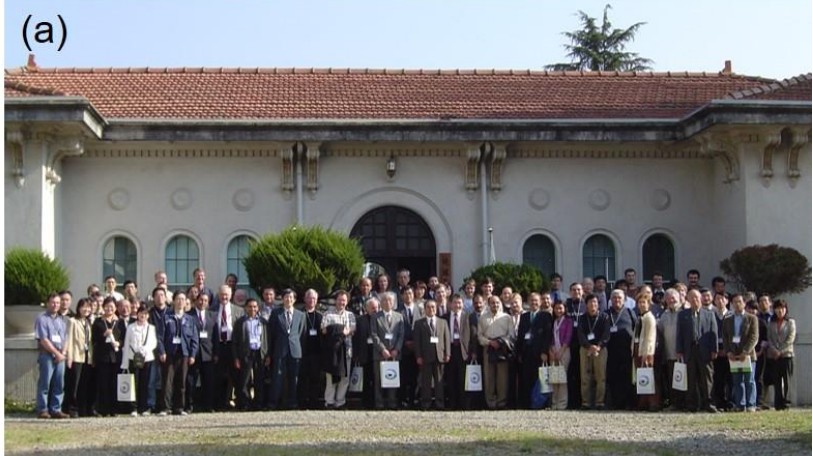

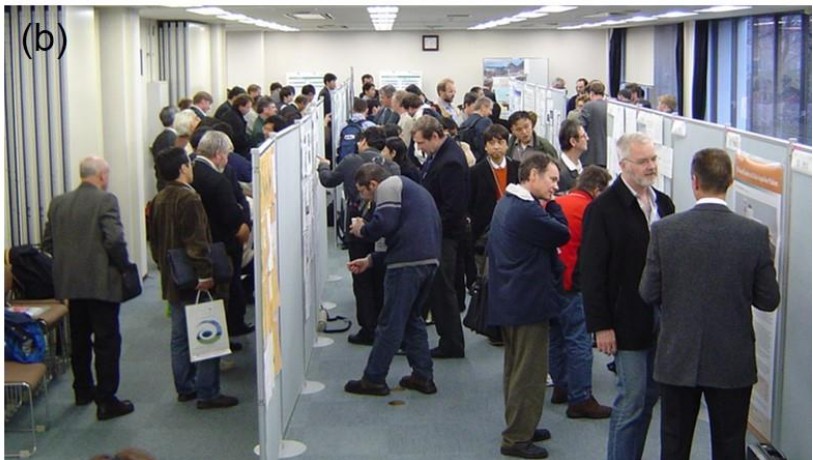

**Figure 30: The 11th IAGA Workshop on Geomagnetic Observatory Instruments, Data Acquisition and Processing held on Nov.9-17, 2004. The measurement session was held at Kakioka (a), while the scientific session was held at Tsukuba (b) (photo supplied by Kakioka Magnetic Observatory).**

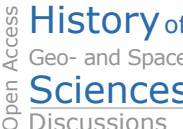

As JMA proceeded with a plan of reduction of its workforce, KMO had to reorganize its observations after the Workshop. At first KMO made an investment in its equipment. The artificial geomagnetic disturbances monitoring system was developed in 2007 to detect and estimate noises caused by magnetic bodies like cars near the property of the Observatory (Minamoto et al., 2011; Nagamachi et al., 2013). The system consists of magnetometers and web cameras and can be

operated remotely. In 2010-2011, fluxgate magnetometers at KAK, MMB, and KNY were replaced with high sensitivity ones (Yamazaki and Mishima, 2013) and data acquisition systems of the geomagnetic field and geoelectric field at KAK, MMB, and KNY were updated so that the 0.1 second values of the both fields were free from system filters. After those preparations, KMO unmanned MMB and KNY in April, 2011. The observation of the atmospheric electric field at MMB was terminated when the observatory was unmanned. Most of the MMB and KNY staff moved to KAK and one staff

member for each MMB and KNY works for a closest Meteorological Observatory in order to conduct the absolute measurement once two weeks.

As the final process of unmanning MMB and KNY was going on, KAK was struck by an unexpected disaster. The 2011 Tohoku Earthquake on March 11, 2011 damaged some buildings and instruments at KAK where is more than 200km southwest from the epicentre area. The area around KAK suffered from failures of electric power and water supply and the

kerosene and petrol were sold out in a series of aftershocks. The KAK staff struggled to keep their magnetic and electric observations going in a confusion. The emergency power supply at KAK was used to make the observations run and the staff made every effort to get the kerosene for the emergency power. Fallen instruments were re-installed and broken ones were temporarily fixed. As a result, KMO provided the geomagnetic and geoelectric field data without data gaps and effects of moves of the fluxgate magnetometer with tremors were later corrected by using the level meter record. The atmospheric

electric field observation by the water dropper was resumed on March 14, only 3 days after the main shock and 1 day before a small explosion of Fukushima Daiichi nuclear power plant. Those data contributed to investigations on the geomagnetic field variation due to the 2011 Tohoku Earthquake (e.g., Utada et al., 2011) and on the atmospheric electric field variation due to the nuclear accident of Fukushima Daiichi nuclear plant (e.g., Takeda et al., 2011).

As damages by the Tohoku Earthquake were gradually healed, KMO marked its 100th anniversary in 2013. In 2012, a

special session "100 years of geomagnetic observations at Kakioka – contributions to centennial progress of geophysics" was held during the annual meeting of Japan Geoscience Union on May 24 at Makuhari, Chiba (Fig.31). In total 21 talks and 4 posters were given including 5 invited ones. In 2013, Dr. Yoshikawa, Director at that time, gave a commemorative lecture at the city hall of Kakioka in January. The Conductivity Anomaly Research Group held its annual meeting at Kakioka in January to commemorate KMO's 100th anniversary. Dr.Kawamura, the 7th Director, and Prof. Schultz of Oregon State

University gave invited talks and Emeritus Prof. Yukutake of the University of Tokyo reviewed works on the secular variation by Dr.Imamichi, the first director (Yukutake, 2013). In total, 34 talks and 33 posters were given in two days.

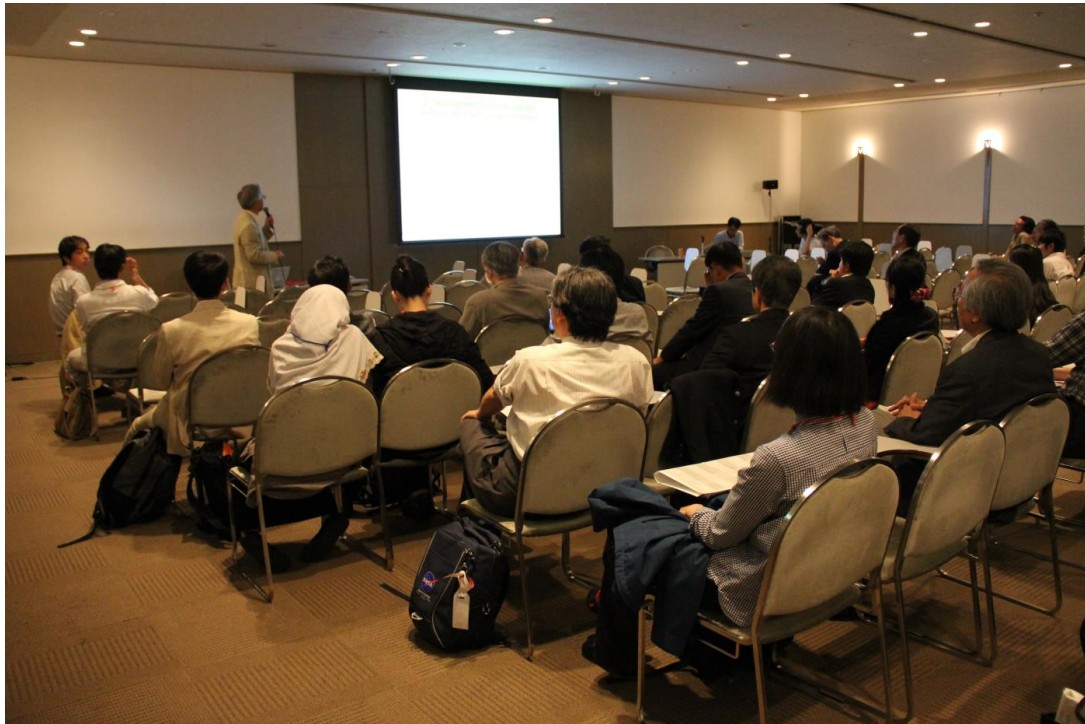

**Figure 31: a special session "100 years of geomagnetic observations at Kakioka – contributions to centennial progress of geophysics" at the annual meeting of Japan Geoscience Union on May 24 at Makuhari, Chiba (photo supplied by Kakioka Magnetic Observatory).**

KMO faced another downsizing in 2021. The observations of the geoelectric field at KAK, MMB and KNY and the atmospheric electric field at KAK were terminated in February, 2021 to follow the government's reform policy. Thus, the history of the electric field observation by JMA and KMO since late 19th century came to its end. The total force observations at Iwaki and Kitaura were also terminated. As a result, KMO has 29 staff members and operates the geomagnetic field observation at KAK, MMB, KNY, and CBI and the total force observation at several volcanos in 2021.

KMO recently has put efforts into two works especially. First, a project to digitize their analogue data has been promoted (Mashiko et al., 2013). Three vector components of the geomagnetic field on photographic papers were digitized every 7.5 sec and 1 min averaged values were computed at KAK back to 1956. As for MMB and KNY, the digitization back to 1971 is done for now. Those digitized data are available via KMO's HP (http://www.kakioka-jma.go.jp/obsdata/metadata/en). They plan to digitize older data and then the analogue data of the geoelectric and atmospheric electric fields would be processed. Second, KMO has cooperated with JMA in monitoring volcanos by the geomagnetic field variation. After the phreatic eruption of Mt. Ontake in 2014, the cooperation was reinforced so that the Overhauser magnetometers were newly installed at 4 volcanoes (Mt. Tarumae, Mt. Azuma, Mt. Ontake, Mt. Kirishima) by JMA to monitor changes of hydrothermal systems inside the volcanos. The total force data which KMO and JMA have accumulated at Japanese volcanoes are revealing conditions inside the volcanos (Hashimoto et al., 2019).

**Data availability**

No data were used in this article.

**Author contribution**

IF designed the study and wrote the manuscript. SN prepared several photographs and searched some information on old measurements.

**Competing interests**

The authors declare that there is no conflict of interest.

**Acknowledgements**

The authors thank the Kakioka Magnetic Observatory for their kind cooperation to this article. IF also thank the editor Kristian Schlegel for his passion and patience.

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
