# Peer review of "History of Kakioka Magnetic Observatory"

_History of Geo- and Space Sciences, 2022_

## Referee Comment (RC1)

Comments on a paper by Fujii and Nagamachi entitled "History of Kakioka Magnetic Observatory" (hgss-2022-5)

This paper describes the history of Kakioka Magnetic Observatory (KMO) from its origin to current status in detail. In particular, both its brilliant and difficult times are introduced under various Japanese social situation or governmental policy changes. The history of instrumentation and staff and their relation to KMO's development are also introduced concisely. The difficulties which KMO has been experienced and struggled described in this paper seem to be more or less common for many geomagnetic observatories in the world. This paper is not only valuable as a description of the history of a geomagnetic observatory but also useful for the society of geomagnetism in the world. Therefore, this paper is worth to be published.

Following is some minor points to be corrected or considered.

1. Page 3, lines 7-8:
   The field notes by Tadataka Ino describes the direction of the target (mountain etc.) measured by his compass without the knowledge on the existence of declination. We can estimate (recover) the declination at that time from his measurement with current geographical information, and hence, his notes themselves are not the record of geomagnetic field. So, the description should be modified. For example, '.... can be estimated ....' etc.

2. Page 3, line 9:
   Kato (1983) is not listed in References.

3. Page 3, Lines 23-24:
   (Hydrographic Bureau, 1983) and (Geological Survey of Japan, 1884, 1885, 1886) are not in References.

4. Page 3, line 34:
   The Geographical Bureau (1886) is not in References.

5. Page 4, line 2:
   'Kkitanomaru' should read 'Kitanomaru'.

6. Page 10, Figure caption:

(b) Wild-Edelmann theodolite and Wild-Edelmann earth inducer --- Which is theodolite and which is earth inducer?

7. Page 12 and page 14:
   If possible, the year when these photos were taken should be given.

8. Page 29, line 17:
   Japan Society for the Promotion of Science (1951)
   -- Which is correct, 1951 or 1950? In the References, it is 1950.

9. Page 40, Line 34:
   'International Magnetosphere Study' should be 'International Magnetospheric Study'.

10. Page 43, line 11:
    The URL where we can read 'Gijyutsu Houkoku' or 'Memoir of Kakioka Magnetic Observatory' should be indicated.

11. Page 45, line 7:
    Kuwashima, 2012 is not listed in References.

12. Page 45, line 7:
    'was determined' should be 'decided'?

13. Page 53, line 14:
    The derivation of the Dst index at WDC for Geomagnetism, Kyoto started in 1986 (not 1973).

14. Page 54, line 11:
    The former English name of SGEPSS is 'Society of Terrestrial Magnetism and Geoelectricity of Japan'.

15. Page 56, line 9:
    (Wakabayashi, 2019) is not in References.

16. Page 59, line 16:
    (Okada et al., 2005) is not in References.

17. page 67, line 23-24:

   The Geophysical Breau .. is not referred in the text. Or is this 'Geographical Bureau'?

18. Page 68, line 3:

 Wadati and Kuwano, 1938 is nor referred in the text.

19 Page 64-69, References:

 At the end of many references, ',last accessed 18, April 2022' is written. These should be removed.

---

## Author Comment (AC1)

Response to comments from Reviewer 1

We greatly appreciate Reviewer 1's comments and support to our work. We basically agree his/her suggestions and applied corrections pointed out. Out responses are given below.

*1. Page 3, lines 7-8:*
*The field notes by Tadataka Ino describes the direction of the target (mountain etc.) measured by his compass without the knowledge on the existence of declination. We can estimate (recover) the declination at that time from his measurement with current geographical information, and hence, his notes themselves are not the record of geomagnetic field. So, the description should be modified. For example, '.... can be estimated ....' etc.*

The sentence is modified as '… can be known …'.

*2. Page 3, line 9:*
*Kato (1983) is not listed in References.*

The reference is as follows.
Kato, A.: The magnetic compass of Masamune Date and secular variations of the declination around Japan, in Centennial history of geomagnetic observations in Japan, Ed. by Kakioka Magnetic Observatory, 101-108, 1983

*3. Page 3, Lines 23-24:*
*(Hydrographic Bureau, 1983) and (Geological Survey of Japan, 1884, 1885, 1886) are not in References.*

The references are added with some changes. Hydrographic Bureau (1983) should be Hydrographic Bureau (1883) as follows.

Hydrographic Bureau: Report of the Naval Observatory, 76pp.,1883 (in Japanese).

We could not access the original references (Geological Survey of Japan in Report of Minister of Agriculture and Commerce, 1884, 1885, 1886), therefore we refer to a work which refers to them as follows.

Supporting Group for Centennial Anniversary of Geological Survey of Japan: Centennial history of

Geological Survey of Japan, Ed. by Editorial Committee of Centennial History of Geological Survey of Japan, Tsukuba, 162pp., 1983 (in Japanese).

*4. Page 3, line 34:*
*The Geographical Bureau (1886) is not in References.*

The name of the bureau was mistakenly written as 'Geophysical' in References. It was corrected.

*5. Page 4, line 2:*
*'Kkitanomaru' should read 'Kitanomaru'.*

   Actually, we realized this should be 'Honmaru' not 'Kitanomaru'. Mistakes were corrected.

*6. Page 10, Figure caption:*
*(b) Wild-Edelmann theodolite and Wild-Edelmann earth inducer---Which is theodolite and which is earth inducer?*

The figure caption of Fig.5 was changed to specify two equipments as '… (b) Wild-Edelmann theodolite (right) and Edelmann earth inductor (center)'.

*7. Page 12 and page 14:*
*If possible, the year when these photos were taken should be given.*

The photo of Fig. 7 was taken in 1927. It was added in the figure caption. As for Fig.6, the year it was taken is unknown.

*8. Page 29, line 17:*
*Japan Society for the Promotion of Science (1951)*
*--Which is correct, 1951 or 1950? In the References, it is 1950.*

Year 1950 is correct. The year in the text was corrected.

*9. Page 40, Line 34:*
*'International Magnetosphere Study' should be 'International Magnetospheric Study'.*

   The spelling of the word was corrected.

*10. Page 43, line 11:*
*The URL where we can read 'Gijyutsu Houkoku' or 'Memoir of Kakioka Magnetic Observatory'*
*should be indicated.*

The URL for Memoir of Kakioka Magnetic Observatory was added to Page 29 where the journal is firstly mentioned. Unfortunately, Gijyutsu Houkoku is available only off-line and a sentence was added to explain it in Page 43. The URL for Technical Report of Kakioka Magnetic Observatory is also added in Page 59.

*11. Page 45, line 7:*
*Kuwashima, 2012 is not listed in References.*

The reference is added as follows.
Kuwashima, M.; Obituary of Dr. Kazuo Yanagihara, Transactions of Society of Geomagnetism and Earth, Planetary and Space Sciences, 212, 15-16, 2012 (in Japanese).

*12. Page 45, line 7:*
*'was determined' should be 'decided'?*

Descriptions by many his colleagues give us the impression that Dr. Yanagihara was a determined person and that was a driving force to develop KASSMER. Therefore, we made no change in this part.

*13. Page 53, line 14:*
*The derivation of the Dst index at WDC for Geomagnetism, Kyoto started in 1986 (not 1973).*

Thank you for pointing out this mistake. It was corrected.

*14. Page 54, line 11:*
*The former English name of SGEPSS is 'Society of Terrestrial Magnetism and Geoelectricity of*
*Japan'.*

It was corrected.

*15. Page 56, line 9:*
*(Wakabayashi, 2019) is not in References.*

The reference is added as follows.

Wakabayashi, Y.: Study of operational meteorology in Japan, University of Tokyo Press, Tokyo, 384pp., 2019 (in Japanese).

*16. Page 59, line 16:*
*(Okada et al., 2005) is not in References.*

The reference is as follows.
Okada, M., Toya, T., Koike, K., Owada, T., Nakajima, S., Shigeno, N., Muromatsu, F., Ookawa, T., Tokumoto, T., Imaizumi, T., Tanaka, T., Sawada, M., Iwase, Y., Ikoma, Y., Kaito, M., Koike, T., Akutagawa, M., Kumasaka, N., Kameya, A., Uesugi, T., Akashi, T., Takahashi, H., Hasegawa, H., Ishida, N., Yokoyama, M., Yamagishi, K., Akita, Y., Kumagaya, N., Iwakata, H., Ose, M., Koide, T., Ishii, Y., and Fujii, I.: Reports on the XIth IAGA Workshop on Geomagnetic Observatory Instruments, Data Acquisition and Processing held at Kakioka/Tsukuba, Japan, in 2004, Tec. Rep. Kakioka Mag. Obs., 3, 1-62, 2005.

*17. page 67, line 23-24:*
*The Geophysical Breau .. is not referred in the text. Or is this 'Geographical Bureau'?*

'Geographical Bureau' is correct. The correction was made.

*18. Page 68, line 3:*
*Wadati and Kuwano, 1938 is nor referred in the text.*

That reference is removed. In addition, 'Wadati' was missing when Wadati (1927) was referred in Page 22, so 'Wadati' was added there.

*19 Page 64-69, References:*
*At the end of many references, ',last accessed 18, April 2022' is written. These should be removed.*

When an URL is shown to access to a reference, we add the date we accessed it as following HGSS's web page reference format. We accessed many references again just before our submission to make sure the URLs are active, so the same date repeatedly appears in References. The format of the date was modified so that it follows HGSS's format precisely.
  Most of URLs in References are for web pages KMO opens their publications through their home

page. Those publications are not tied to the DOI system yet, therefore giving URL might be useful to readers. However, we understand those descriptions may be removed or modified in the HGSS's checking process before publishment.

---

## Author Comment (AC2)

Response to comments from Reviewer 2 (Dr. Adrian Hitchman)

We appreciate Reviewer 2's comments and support to our work. We are also very grateful to her careful reading of the text written in a poor English. We basically accept her suggestions about the language. Out responses are given below.

*While I am unable to record in detail the numerous minor edits that would polish the text, I do recommend the following more significant amendments:*

*In different places in the text, replace*

1. *"manned" with "staffed" and "unmanned" with "unstaffed" to use gender neutral text where possible*
2. *"head quarter" and "headquarter" with "headquarters"*
3. *"cupper" with "copper"*
4. *"alternative current" with "alternating current"*

Replacement was done as suggested.

*page 1, line 3: does the Meteorological College postcode have some numbers missing? – "270-????"*

As the reviewer suspected, the postcode of Meteorological College's address was missing. It was completed as 277-0852.

*p 1, l 25: correct the spelling of "Kanoya"*

It was corrected as 'Kanoya'.

*p 2, l 9: replace "war load" with "warlord"*

It was corrected as suggested.

*p 8, l 1: replace "dart-covered" with "dirt-covered" (I think)*

The reviewer is correct. The mistake was corrected.

*p 18, l 6: replace "Verlin" with "Berlin"*

It was corrected as suggested.

*p 22, l 8: replace "consigned" with "assigned"*

It was corrected as suggested.

*p 26, l 4: replace "electorode" with "electrode"*

It was corrected as suggested.

*p 31, l 10: replace "function" with "operate"*

It was changed as suggested.

*p 45, l 1: change the Section number from "8" to "9", and then add 1 to all later Section number*

Thank you for spotting this embarrassing mistake. The section is numbered correctly now.

*p 54, l 1: "booming economy" is more commonly used (although "blooming economy" works too)*

We use 'booming economy'.

*p 57, l 2: replace "island" with "Island"*

It was corrected as suggested.

---

## Author Response (AR1)

Response to comments from Topical Editor, Dr. Kusumita Arora

We greatly appreciate Dr. Arora's comments and careful checks of our manuscript. We agree most of suggestions and applied corrections for English mistakes the Editor pointed out. Also, we found two mistakes and corrected them (See 10 and 11 below). Together with corrections made by Reviewers' suggestions, we prepared revised manuscripts with and without the corrections marked.

Out responses to the Editor's comments are listed below.

*1. Abstract, the bottom line:*
's' was added to verbs as KMO is supposed to be a single organization.

*2. Page 1, the bottom line:*
 'HP' was replaced with 'website'.

*3. Page 2, Figure 1 caption:*
 The font for 'red circles' were changed into the one that was used in the other parts.

*4. Page 3, the first line:*
 The word 'article' was replaced with 'articles'.

*5. Page 3, line 11:*
 The word 'temporal' was replaced with 'short-lived'.

*6. Page 3, Lme 30:*
We think the first sentence of this paragraph will be of help for readers to understand that the main topic of the paragraph is 'the reason of three independent responses to the First International Polar Year'. Therefore the sentence was not removed.

*7. Page 31, Lme 30:*
We mean that Dr.Imamichi continued his job after the war. Therefore, we leave the sentence as it is.

*8. Page 40, line 32:*
Looking at several webpages of research institutions, IQSY seems to have two names as International Quiet Sun Years and International Years of Quiet Sun. The abbreviation IQSY is more suitable for International Quiet Sun Years, so the phrase is left as it is.

*9. Page 41, Line 2:*

It should be IASY not IQSY. So, it is left as it is.

*10. Page 53, line 19:*

Computer names were wrong. So they were corrected as 'updated from HITAC-10 to HITAC E-600'.

*11. Page 53, lines 21 ans 22:*

We found that the 3 second values and 1 second values were only mentioned in the 1983-1984 yearbooks. So we delete the sentence about publishment of the 3 second values and 1 second values.

*12. mistakes*

Following the Editor's suggestions, grammatical mistakes and so on were corrected as follows.

Page 3, Line 4: 'this article' -> 'the present article'.

Page 3, Line 5 and many others: '20th century' -> 'the 20th century '.

Page 3, Line 10: 'war load' -> 'warlord '.

Page 3, Lines 13-14: 'in order to anticipate the observation activity' -> 'in anticipation'

Page 3, Line 15 and many others: 'The' -> 'the'.

Page 3, Line 16: 'participate' -> 'participate in'.

Page 3, Line 16: 'Then, with some reason' -> 'Later'.

Page 3, Line 24 and many others: 'Hydrographic Bureau' -> 'the Hydrographic Bureau'.

Page 3, Line 26: 'were transferred and suspended in several times' -> 'had several interruptions'

Page 3, Line 30: '14 years passed' -> '14 years had passed'

Page 3, Line 31: 'came to the end' -> 'had come to an end'

Page 3, Line 32 and many others: 'a' -> 'the'

Page 3, Line 32: 'suspected' -> 'speculated'

Page 3, Line 33: 'competing each other' -> ' competing with each other'

Page 6, Line 1: 'engaged to' -> 'engaged with'

Page 6, Lines 4 and 5: 'on the data books' -> 'in the data books'

Page 6, Line 8: 'grew as' -> 'grew into'

Page 6, Line 16: 'where was likely to be off future railway' -> 'where development of future railways was unlikely'

Page 6, Lines 17 and 18: 'They favored these geological settings of Kakioka' -> 'The geological settings of Kakioka was preferred'

Page 11, Line 6: 'it couldn't help losing' -> 'it lost'

Page 22, Line 9: 'might help its operation' -> 'might have helped in its operation'

Page 23, Line 1: 'Second polar Year' -> 'Second Polar Year'

Page 25, Line 2: 'Cupper' -> 'Copper'

Page 25, Line 2: 'deep' -> 'depth'

Page 25, Line 10: '1940's' -> '1940's are available'

Page 27, Line 6: 'are' -> 'have'

Page 29, Line 1: 'the staff' -> 'staff'

Page 29, Line 13: 'were gradually faded' -> 'gradually faded'

Page 29, Line 15: 'circumstance' -> ' circumstances'

Page 29, Line 22: 'Institution' -> ' institution'

Page 29, Line 24: 'subtle' -> ' rare'

Page 31, Line 3: 'at' -> 'during'

Page 31, Line 6: 'be' -> 'leave it'

Page 35, Line 1: 'proceeded, too' -> also intiated'

Page 40, Line 4: 'at Kakioka. Most' -> at Kakioka as most'

Page 40, Lines 4 and 5: 'most of section locate within 30km from Kakioka' -> 'most of section was located within 30km of Kakioka'

Page 40, Line 15: 'afterwards' -> 'later'

Page 40, Line 21: 'the vicinity of Kakioka remained rural' -> 'weaker urbanization of the area around Kakioka'

Page 40, Line 25: 'prompted' -> 'promoted'

Page 40, Lines 35 and 36: 'verified their observation to adapt to the activities. Two big changes brought in observations at KMO in 1950's– 1960's would be ' -> 'two big changes were brought in to observations at KMO in 1950's – 1960's:'

Page 41, Line 4: 'at the first time' -> 'for the first time'

Page 45, Line 2: 'Both of' -> 'Both'

Page 45, Line 7: 'use' -> 'using'

Page 49, Line 4: 'inputted' -> 'uploaded'

Page 53, Line 11: 'and then the economy turned booming' -> 'before the economy recovered and turned booming'

Page 49, Line 38: 'Participation to' -> 'Participation in'

Page 54, Line 1: 'blooming' -> 'booming'

Page 56, Line 25: 'fist' -> 'first'

Page 56, Line 29: 'Namely' -> 'Basically'

Page 56, Line 30: 'way' -> 'nature'

Page 56, Line 36: 'apply' -> 'apply for'

Page 59, Line 8: '21th' -> '21st'

Page 59, Line 10: 'there' -> 'its'

Page 61, Line 7: 'the both fields' -> 'both the fields'

Page 61, Line 13: 'where' -> 'which'

Page 61, Lines 27 and 31: 'given' -> 'presented'